# A moonlighting role for enzymes of glycolysis in the co-localization of mitochondria and chloroplasts

Youjun Zhang [1,2✉], Arun Sampathkumar [1], Sandra Mae-Lin Kerber [1], Corné Swart[1], Carsten Hille [3,5], Kumar Seerangan[1], Alexander Graf[1], Lee Sweetlove [4] & Alisdair R. Fernie [1,2✉]

Glycolysis is one of the primordial pathways of metabolism, playing a pivotal role in energy metabolism and biosynthesis. Glycolytic enzymes are known to form transient multi-enzyme assemblies. Here we examine the wider protein-protein interactions of plant glycolytic enzymes and reveal a moonlighting role for specific glycolytic enzymes in mediating the co-localization of mitochondria and chloroplasts. Knockout mutation of phosphoglycerate mutase or enolase resulted in a significantly reduced association of the two organelles. We provide evidence that phosphoglycerate mutase and enolase form a substrate-channelling metabolon which is part of a larger complex of proteins including pyruvate kinase. These results alongside a range of genetic complementation experiments are discussed in the context of our current understanding of chloroplast-mitochondrial interactions within photosynthetic eukaryotes.

[1] Max-Planck-Institut für Molekulare Pflanzenphysiologie, Am Mühlenberg 1, 14476 Potsdam-Golm, Germany. [2] Center of Plant Systems Biology and Biotechnology, 4000 Plovdiv, Bulgaria. [3] Department of Physical Chemistry, University of Potsdam, Karl-Liebknecht-Strasse 24-25, 14476 Potsdam-Golm, Germany. [4] Department of Plant Sciences, University of Oxford, South Parks Road, Oxford OX1 3RB, UK. [5] Present address: Technical University of Applied Sciences Wildau, Hochschulring 1, 15745 Wildau, Germany. ✉email: Yozhang@mpimp-golm.mpg.de; fernie@mpimp-golm.mpg.de

Glycolysis represents one of the hallmark pathways of respiration, providing carbon skeletons for the biosynthesis of a wide range of metabolites as well as being at the heart of energy transformations[1,2]. Glycolytic enzymes are known to form multienzyme complexes in a broad range of species, yet the function of these assemblies remains unclear[3,4]. Recent studies in yeast[5], blood cells[6], and plants[7] demonstrate that glycolytic enzymes colocalize to areas on membranes where ATP is rapidly consumed, suggesting a regulatory role of such enzyme assembles[3,4]. The dynamics of glycolytic enzyme assemblies were recently followed using fluorescence resonance energy transfer (FRET)[8] and fluorescence recovery after photobleaching in living human cells[9], demonstrating that the first four enzymes of the glycolytic pathway each independently follows its own specific substrate gradient—providing an important insight into the potential mechanism of assembly of enzyme complexes[4,9,10]. Moreover, using fluorophore-tagged enzymes to follow movement in microfluidic devices, enzymes have been suggested to show chemotactic movement along their substrate gradient[11]. However, it is important to note that these studies remain somewhat controversial, with the argument being raised that the observations may be an artifact of the measurement method[11].

In plants, as in microbes and mammals, glycolytic enzymes have been demonstrated to be physically associated with mitochondria[12]. Further important experiments revealed that the mitochondrially associated enzymes were capable of sustaining respiratory flux when supplied with isotopically labeled glucose[12] and that the association of the enzymes with mitochondria was dynamic and dependent on the respiratory demand of the cell[4,7]. Despite the initial work being some 15 years old[7,12], and in contrast to the well-characterized interactome of the plant TCA cycle (tricarboxylic acid cycle)[13,14], glycolytic assemblies have not been well characterized in plants. Here, we set out to perform a comprehensive characterization of these interactions. Serendipitously, we discovered evidence for a moonlighting role of these enzymes in bringing chloroplasts and mitochondria together.

The association of organelles has previously been observed and is often postulated to aid metabolic pathways that span organelles, such as photorespiration and nitrogen assimilation, and may also be beneficial for biochemical efficiency[15–18]. In addition to the clear metabolic intimacy of the organelles of the plant cell, mitochondria and chloroplasts have frequently been observed to spatially colocalize—a feature that has, among other considerations, been hypothesized to be important to ensure energy-use efficiency[16,19,20]. The direct interaction between mitochondrial and chloroplast outer membranes has been demonstrated using transmission electron micrographs (TEMs) of parenchyma cells from *Citharexylum myrianthum*[21]. It has also become apparent that membrane microdomains may be important for organelle trafficking, cellular signaling, and organelle tethering[16,19], and it has been speculated that metabolite exchange between organelles could be regulated by these microdomains, which could involve direct channeling of molecules and synchronization without the necessity for cytosolic involvement[15,18,19]. Following this theory, the exchange of metabolites between respiration and photosynthesis would be more efficient. However, it remains unclear how spatial colocalization of plant organelles is achieved or regulated.

In the current study, we used contemporary molecular and cell biological techniques to study the interaction partners of the cytosolically localized glycolytic enzymes. Having identified robust protein–protein associations between phosphoglycerate mutase (PGAM), enolase, and pyruvate kinase, we demonstrated that 2-phosphoglyceric acid (2PGA), but not phosphoenolpyruvate (PEP), was fully channeled in the subset of the glycolytic enzymes, which were bound to the mitochondria. Finally, we utilized mutants of these enzymes to gain a first insight into the physiological role of this complex identifying a novel moonlighting role for these proteins in the colocalization of chloroplasts and mitochondria, which was independent of their catalytic activity.

## Results

**Construction of the plant glycolytic interactome.** Thirty-seven cytosolic and mitochondrial outer-membrane proteins of *Arabidopsis thaliana*—including all constitutively expressed subunits of the enzymes of glycolysis as well as membrane proteins—have proven to mediate the transport of triose phosphates (including 3-phosphoglycerate; 3PGA), and pyruvates were tested for binary protein–protein interactions (Supplementary Data 1)[22]. Four quantitative assays were employed to gain a reliable binary protein interaction map of plant glycolysis. Affinity purification-mass spectrometry (AP-MS) assays were conducted using the PSB-D *Arabidopsis* cell culture line (Fig. 1a)[23]. Interactions between subunits in known protein complexes, including phosphofructokinase (PFK), fructose-bisphosphate aldolase (ALD), and pyruvate kinase complex, were well captured by this approach (Fig. 1a)[7,23].

Most of the glycolytic genes appeared to be dually localized within the cytosol and associated with mitochondria according to both experimental evaluation of the green fluorescent protein (GFP) signal of AP-MS using confocal microscopy and based on the SUBA4 proteomics database[23,24] (Fig. 1b). In addition, bimolecular-fluorescence complementation (BiFC) assays in *Arabidopsis* leaves and mesophyll protoplasts, which enabled qualitative but highly sensitive detection of protein–protein interactions, were employed for binary protein interaction analysis (Fig. 1c–i and Supplementary Figs. 1 and 2)[25]. Reversible binary protein interaction assays based on split luciferase (split-LUC) in *Arabidopsis* mesophyll protoplasts were employed to verify these protein interactions (Fig. 1j). Finally, coimmunoprecipitation (Co-IP) was carried out using mCherry tag-based bait in *Arabidopsis* leaves in order to evaluate dual protein interaction of each by western blotting (Supplementary Fig. 1b). It is important to note that these approaches are based on different principles and were performed in widely different physiological conditions. As such, it can be anticipated that complementary results from all four approaches will identify genuine interacting protein pairs. In addition to the complexes detected previously, a new complex was identified between PGAM1, enolase, and pyruvate kinase, which were all dually associated with mitochondria as well as occurring free within the cytosol (Fig. 1 and Supplementary Fig. 2).

The binary interactions between the novel complex and the membrane proteins of the chloroplast triose phosphate translocator (TPT) and mitochondria voltage-dependent anion channel (VDAC) were next tested by BiFC (Fig. 1c–i) and split-LUC (Fig. 1j) assays[25]. The TPT transiently interacted with PGAM in the BiFC and split-LUC assays, but no stable interactions were obtained between TPT and PGAM by AP-MS, suggesting an indirect interaction consistent with the TPT residues in the inner chloroplast membrane[22]. By contrast, substantiated interactions within phosphoglycerate mutase, enolase, and pyruvate kinase complex, and between pyruvate kinase and VDAC were underscored by all methods tested. VDAC, located at the interface between the cytosolic and mitochondrial environments, could play an important role in regulating metabolite flux through the outer mitochondrial membrane[7]. In addition, the interaction of pyruvate kinase 4 (PYK4) and VDAC might relate to the import of pyruvate into the mitochondria, providing a potential biochemical role for the associations.

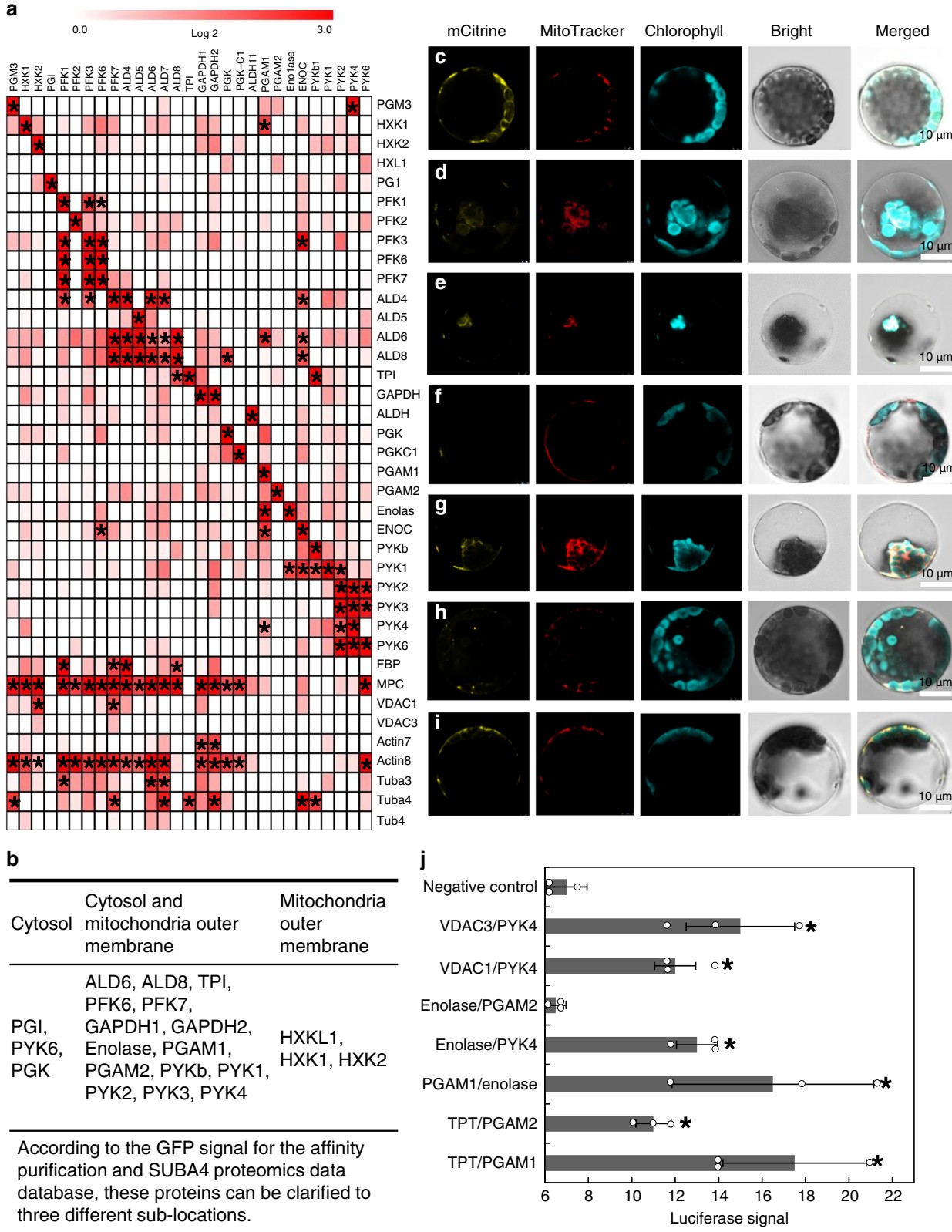

**In vivo analysis of the novel glycolytic complex**. Identifying the correct subcellular locations for all enzymes and metabolites in plant metabolic networks has proven to be critically important for the success of the new generation of large-scale metabolic models that are driving a network-level appreciation of metabolic behavior[26]. We have previously shown that several glycolytic enzymes interact with the surface of the mitochondria[7]. Similarly, proteomic analyses of isolated mitochondria have consistently demonstrated the association of glycolytic enzymes (Fig. 1b). For example, PGAM1 is colocalized to the cytosol and the outer membranes of both mitochondria and chloroplasts (Fig. 2a, b). Enolase is colocalized to both the cytosol and the outer membranes of the mitochondria, while the alternative translation product of enolase (AtMBP-1) was also localized to the nucleus (Fig. 2b–d)[27]. Most of

**Fig. 1 The protein–protein interaction of *Arabidopsis* glycolytic enzymes. a** The protein interactions calculated fold change in the heatmap. The novel complex of phosphoglycerate mutase 1–enolase–PYK4 complex could be detected. The star mark was represented fold change more than four times compared with GFP control. **b** Sublocation of all the glycolysis according to the SUBA4 database mitochondria proteomics data and GFP signal. **c** NE-TPT/ PGAM1-CE. **d** NE-TPT/PGAM2-CE. **e** Enolase-NE/PGAM1-CE. **f** Enolase-NE/PGAM2-CE. **g** Enolase-NE/PYK4-CE. **h** VDAC1-NE/PYK4-CE. **i** VDAC3-NE/ PYK4-CE. Confirmation of phosphoglycerate mutase 1–enolase–PYK4 complex with membrane protein (chloroplast TPT and mitochondria VDAC) by bimolecular fluorescent complementation (BiFC) assay. Interactions among chloroplast and mitochondria membrane protein and phosphoglycerate mutase 1–enolase–PK4 complex were further tested by BiFC with transient expression of tagged proteins in *Arabidopsis* mesophyll protoplasts. NE is the N-terminal of the split-mCitrine, CE is the C-terminal of the split-mCitrine. The panels from the left side show the BiFC fluorescence, fluorescence from bright-field image, MitoTracker Red staining, autofluorescence, blank, and the merged image of all of those, respectively. **j** Confirmation of protein–protein interaction by split *Renilla* luciferase assay. Interactions among chloroplast and mitochondria membrane protein and phosphoglycerate mutase–enolase–pyruvate kinase complex were further confirmed by split *Renilla* luciferase with transient expression of tagged proteins in *Arabidopsis* mesophyll protoplasts. One-way ANOVA analysis by enolase/GFP as a negative control (*$P < 0.05$, SD). PGM phosphoglucomutase, HXK hexokinase, PGI phosphoglucose isomerase, PFK phosphofructokinase, ALD aldolase, TPI tripsephosphate isomerase, *GAPDH* glyceraldehyde phosphate dehydrogenase, PGK phosphoglycerate kinase, ENO enolase, PYK pyruvate kinase, PGAM phosphoglycerate mutase, TPT triose phosphate translocator, VDAC voltage-dependent anion channel.

the pyruvate kinase subunits were localized to both the outer membranes of the mitochondria and cytosol, while PYK4 frequently interacted with enolase at the outer membrane of the mitochondria (Figs. 1b and 2d–f).

In addition, we used FLIM-FRET (Förster resonance energy transfer measured by fluorescence lifetime imaging microscopy) to evaluate the interactions of pairwise combinations of all target fluorescent fusion proteins in order to gain a more detailed knowledge of the organization of the glycolytic complex (Fig. 2g). The FLIM-FRET results demonstrate that PGAM1, PGAM2, enolase, and PYK4 form homo- and hetero-oligomers with FRET values higher than those of the controls, confirming their strong interaction (Fig. 2g). Furthermore, both the mitochondrial outer-membrane proteins, VDAC1 and VDAC3, were colocalized with PYK4 (Fig. 2e, f) and indeed yielded the highest FLIM-FRET signal (Fig. 2g), suggesting that they formed the tightest association. As autofluorescence of the chloroplast prevents FLIM-FRET measurement of chloroplast-associated proteins, we did not assess the interaction between TPT and PGAM here. In addition to analyzing the binary interactions, we also coexpressed all three proteins in the plant, and analyzed the resultant transformants in vivo combining PGAM and enolase by split-mCitrine and linking PYK4 by mCherry (Fig. 3a). The mCitrine, mCherry, and chlorophyll autofluorescence were co-sublocalized between chloroplast and mitochondria, demonstrating that the protein complex clearly formed an in vivo link between the outer membranes of chloroplasts and mitochondria (Fig. 3).

**Enzyme activity and analysis of substrate channeling.** Given the accumulated evidence of rapid repartitioning of glycolytic enzymes between two pools—one homogeneously cytosolic and the other associated with the surface of the mitochondria[7], it seems likely that the glycolytic enzyme complex has a regulatory function. Here, we used both purified mitochondria and total extra-plastidial protein extracts to evaluate the relative activities of the enzyme complex in comparison to the free enzymes, respectively. For this purpose, either isolated mitochondria or total extra-plastidial extract were incubated with 3PGA, 1 mM ADP, and 1 mM NADH and *Escherichia coli* lactate dehydrogenase and the rate of $NAD^+$ production of the complex/free enzyme was monitored (Fig. 4a). We next analyzed the activities of the two fractions as a function of different 3PGA substrate concentrations (Fig. 4b, c). In addition, we demonstrated that the total enolase protein content was equivalent in the extra-plastidial fraction and the mitochondrial fraction (Fig. 4c). Compared with the extra-plastidial fraction, the mitochondrially associated complex displayed 19 times greater efficiency ($K_{cat}/K_M$) and lower $K_M$ values (Fig. 4b, d), indicating a high affinity and specificity for the substrate as well as a highly efficient conversion of 3PGA to

pyruvate at the outer mitochondrial membrane. This enzyme–enzyme assembly will thus ensure that carbon reaches mitochondrial respiration, while the freely diffusing enzymes would allow the conversion of 3PGA to 2PGA and the use of both in other metabolic pathways, such as amino acid biosynthesis.

Having demonstrated the assembly of the glycolysis complex, we next assessed whether either 2PGA or PEP were channeled within this complex, via the isotope dilution approach[7]. The principle of this experiment is that it measures the effect of adding an unlabeled intermediate to the rate of the conversion of an isotopically labeled pathway substrate to a labeled end product of that pathway. If metabolites are channeled through the pathway, then labeled intermediates do not enter the bulk phase, and the label is therefore not diluted by mixing with the added unlabeled intermediate[14,26]. It is important to note that this is only the case when the enzymes are saturated with substrate. Given that $^{13}C$-labeled 3PGA is not commercially available, we incubated 100 mM $^{13}C$-labeled glucose with recombinant hexokinase, phosphoglucose isomerase, PFK, ALD, triose phosphate isomerase, glyceraldehyde phosphate dehydrogenase, phosphoglycerate kinase, and 200 mM ATP and 100 mM $NAD^+$ for 2 h prior to the isotope dilution experiment (Fig. 4e). Isolated mitochondria were then incubated in $^{13}C$-labeled substrate until the label accumulation in the product reached an isotopic steady state. Then, an unlabeled intermediate—either 2PGA or PEP—was added and the "dilution effect" on the labeling in the product was monitored over time. Interestingly, the ratio of labeled pyruvate was unchanged after adding unlabeled 2PGA (Fig. 4f), indicating a complete channeling of the conversion of 2PGA to pyruvate at the plant outer mitochondrial membrane. In contrast, addition of unlabeled PEP resulted in a dramatic decline in the proportion of pyruvate labeling (Fig. 4g). These results thus suggest that 2PGA, but not PEP, is channeled within the complex.

**Decreased association between mitochondria and chloroplast.** The improved efficiency of novel glycolytic metabolon appears to play an important role in facilitating the exchange of metabolites across the cytosol and through organelle membranes. This supports the theory of cross-talk between the mitochondria and chloroplast via direct physical interaction[28]. In wild-type (WT) plants, mitochondria are reported to interact in a dynamic fashion with chloroplasts[28] (Fig. 5a, Supplementary Fig. 3a, and Supplementary Movie 1). In order to test if such interactions are perturbed in the *enolase*[29] and *pgam* mutants[30], we monitored mitochondria and chloroplast association by expressing a GFP targeted to the mitochondria[31]. Quantitative analysis of confocal micrographs revealed a significant loss of association between the two organelles in the mutant lines (Fig. 5a, d and Supplementary Movies 1, 2, and 3). Irrespective of genotype no apparent

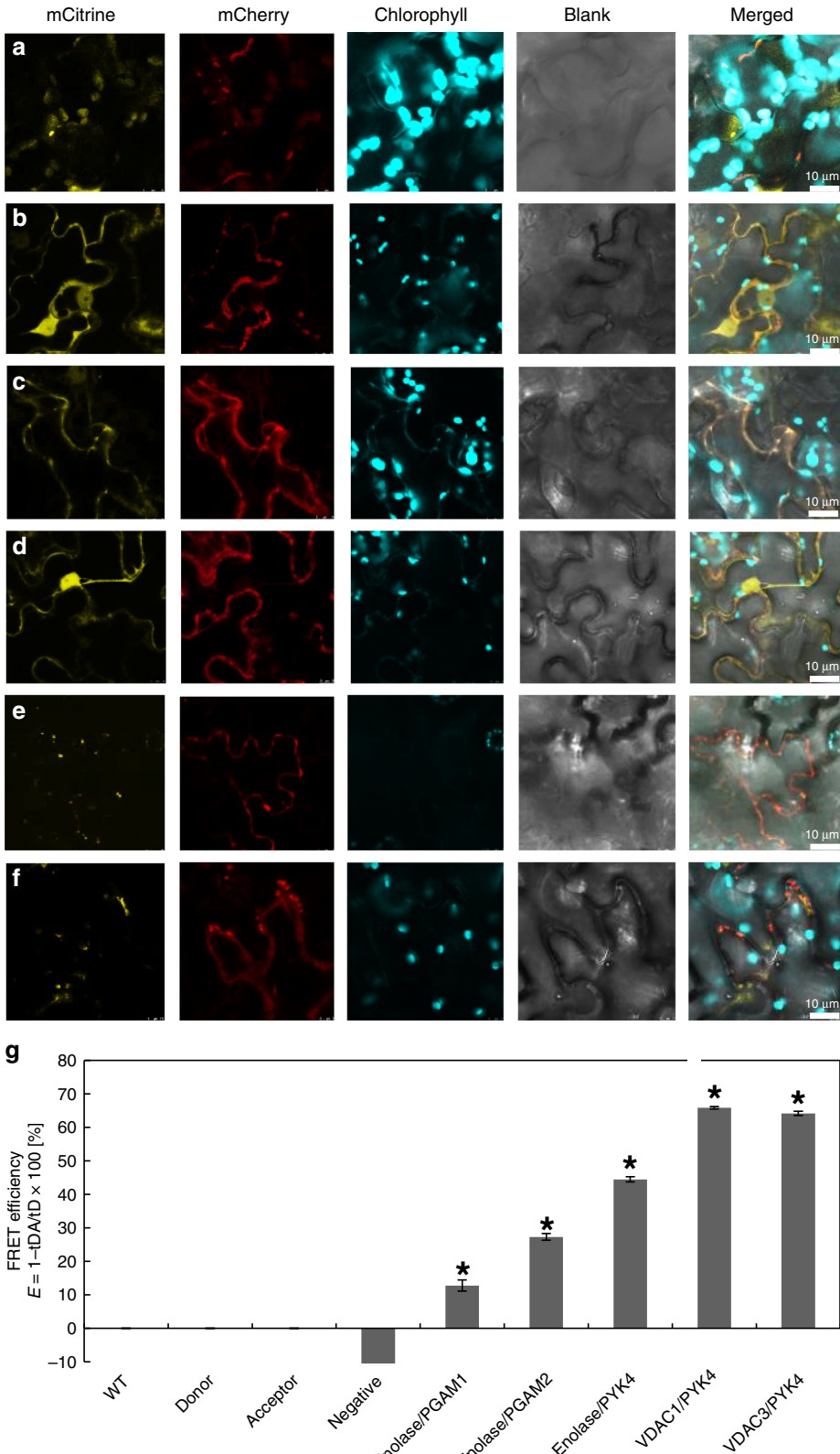

**Fig. 2 The protein–protein interaction among the phosphoglycerate mutase 1–enolase–PK4 complex and VDAC. a** mCitrine-TPT/PGAM1-mcherry. **b** Enolase-mCitrine/PGAM1-mcherry. **c** Enolase-mCitrine/PGAM2-mcherry. **d** Enolase-mCitrine/PYK4-mcherry. **e** VDAC1-mCitrine/PYK4-mcherry. **f** VDAC3-mCitrine/PYK4-mcherry. Co-sublocalization of phosphoglycerate mutase 1–enolase–PYK4 complex with organelles' membrane protein was further tested by confocal with transient expression of tagged proteins in *Arabidopsis* leaves. The panels from the left side show the mCitrine fluorescence, mCherry fluorescence autofluorescence, blank, and the merged image of all of those, respectively. **g** Confirmation of protein–protein interaction by FLIM-FRET assays. All the vectors of the co-sublocalization were transformed into plant cell culture for the FLIM-FRET analysis. The Los2-mcitrine/mCherry was used as negative control (*$P < 0.05$, error bar is SEM with more than 10 replicates).

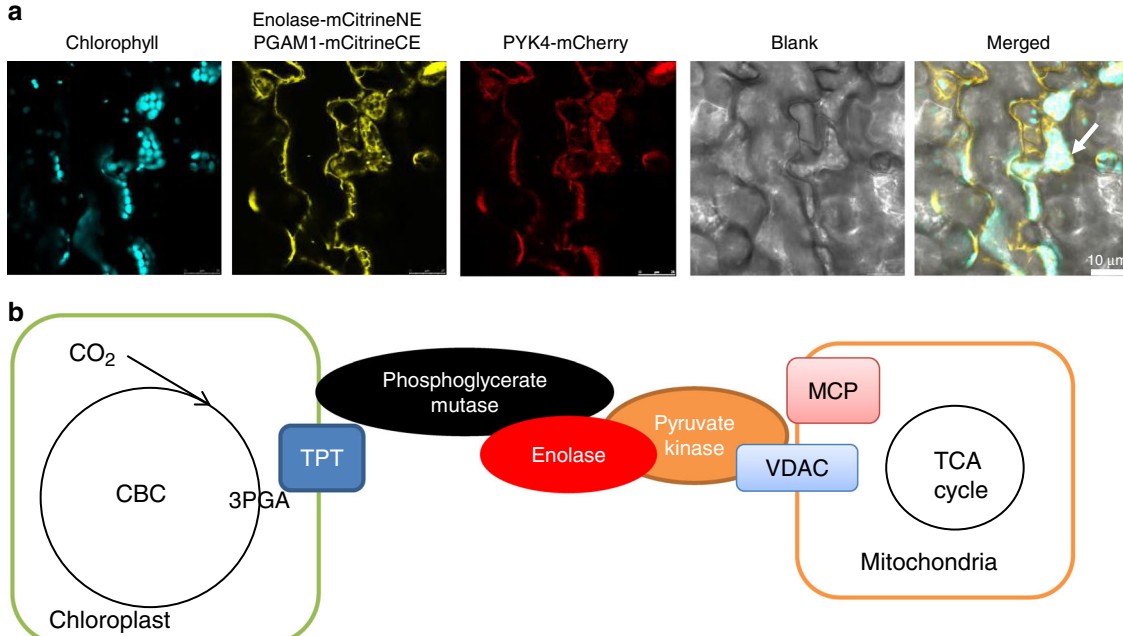

**Fig. 3 The glycolytic enzyme complex between mitochondria and chloroplast in vivo. a** Confirmation of protein complex with two organelles by three protein co-sublocalization. The enolase-mCitrineCE and IPGAM1-mCitrineNE and PYK4-mCherry were coexpressed in the *Arabidopsis* leaves. The novel complex present strong signal between mitochondria and chloroplast. **b** The summary of protein–protein interaction. MPC mitochondrial pyruvate carrier, TPT triose phosphate translocator, VDAC voltage-dependent anion channel, CBC Calvin–Benson–Bassham cycle.

differences were observed between the day and night time points (Fig. 5e, Supplementary Fig. 3) In addition, both mutants had mitochondria that were more clustered together compared to the WT. To further corroborate our results, we quantified such interactions in *enolase* mutants complemented with a nuclear-targeted enolase, which has been previously shown not to fully complement the mutant phenotype[29]. This analysis revealed no significant changes in the subcellular phenotype of these lines (Fig. 5e and Supplementary Fig. 3). To further assess the functional role of the enzymatic activity of PGAM, we visualized such associations in *pgam* mutants complemented with a catalytically inactive version of the enzyme (PGAM1, H39, S80, K360, and H470 mutated). In this case, we observed complete restoration of the association between mitochondria and chloroplast (Fig. 5c and Supplementary Fig. 3c).

To gain deeper insights into the dynamic association of the mitochondria and chloroplast, we captured time-lapse images of mitochondrial movement in the various genotypes lines. This analysis, alongside kymograph-based visualization[32] (Fig. 6), revealed that the mitochondria associate with chloroplasts in a dynamic fashion during which they switch between interacting and non-interacting phases (Fig. 6a–e and Supplementary Movies 1, 2, and 3). Unlike the WT and *sdmA-pgam1/2* mitochondria, which directly interact with and move around chloroplasts (Fig. 6e, Supplementary Fig. 4g, and Supplementary Movie 1), this interaction between mitochondria and chloroplast was not observed in the mutants (Fig. 6f–h and Supplementary Fig. 4g). In addition to rule out the possibility that the plant growth phenotype affects this association, we used the enolase inhibitor (ENOblock hydrochloride, which does not affect the active site of enolase, but rather influences its binding and as such its subcellular localization[33]) to treat lines expressing a mitochondrially targeted GFP under the control of the *UBIQUITIN10* promoter in the Col-0 background (a genotype that displays the same phenotype as WT). Following treatment with ENOblock hydrochloride, both the associations of the mitochondria to chloroplast and the movement of the mitochondria were significantly decreased, while the actin cytoskeleton behavior

remained unaffected (Fig. 7 and Supplementary Fig. 5c). Moreover, based on images of dual localization of mitochondria and peroxisome markers, we do not see significant overlap between the two organelles as compared to the association between mitochondria with chloroplast (Supplementary Fig. 5a, b).

We argue that by quantitatively measuring the displacement of the mitochondria from their chloroplast association, we could understand the dynamics of the association. Interestingly, we observed significantly increased the movement of mitochondria in plants imaged at night compared to the day (Fig. 6f and Supplementary Fig. 4). This is consistent with previous reports of dynamic changes in metabolite levels during different day cycle[34]. Investigation of such dynamics shows that mitochondrial movement is strikingly perturbed in the mutants under both conditions (Fig. 6f and Supplementary Fig. 4). Partial rescue was observed in both the enolase nuclear complementation lines and PGAM-mutated lines (Fig. 6c. f). Full details of the complementation of morphological and cell biological phenotypes were provided in Supplementary Note. Taken together, these results suggest that the glycolytic complex we identified here is highly influential in determining the degree of association between the mitochondria and the chloroplast.

## Discussion

The prevailing view of the cytosol is that it is dominated by random diffusion of enzymes and metabolic intermediates; however, work over the past four decades suggests that dynamic supramolecular assemblies of enzymes are prevalent[35–37]. Such assemblies are often called metabolons since they are assumed to mediate metabolite channeling; however, evidence suggests that the case is indeed relatively scarce and can be difficult to interpret[11,38–42]. The strongest evidence for such studies is provided by the so-called dilution experiments, whereby labeled substrate is provided to a pathway and the labeling pattern in the pathway product is followed over time in response to the provision of unlabeled intermediates. Supportive evidence for substrate channels is, however,

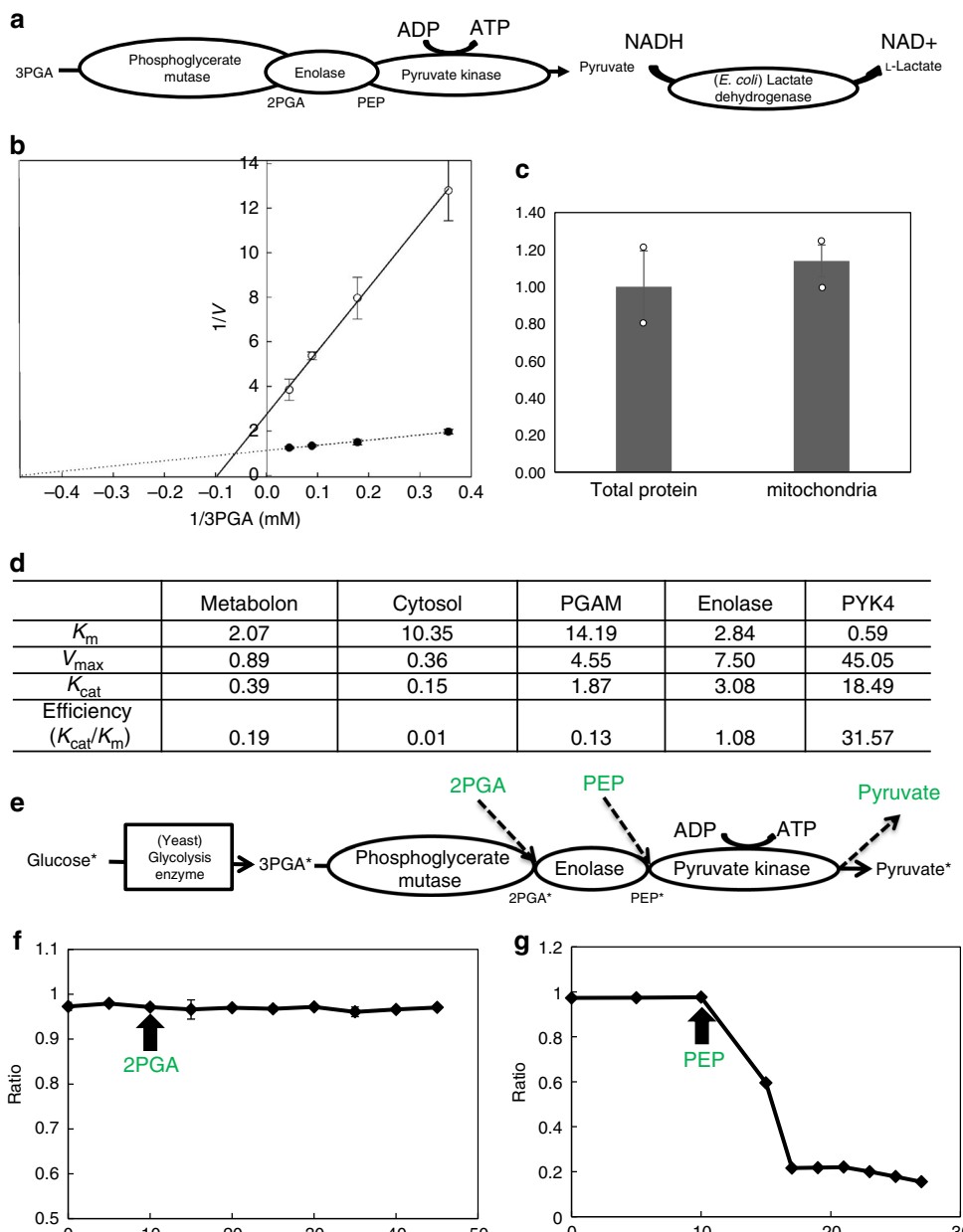

**Fig. 4 The function of the phosphoglycerate mutase–enolase–pyruvate kinase complex in isolated plant mitochondria. a** Mechanism of the enzyme activity measurement. The NADH degradation ratio was measured at OD340 and calculated as $-V$. **b** Using the Michaelis–Menten equation to analyze the protein complex and the glycolytic free enzymes. The substrate concentration that produces a $V_i$ that is one-half of $V_{max}$ is designated the Michaelis–Menten constant, $K_M$. The filled symbols are data from the isolated mitochondria fraction. The open symbols are data from the total extra-plastidial protein extracts. **c** Protein concentration measured by anti-enolase western blotting. Five micrograms of total protein and 5μg of mitochondrial protein were loaded to get the same amount of enolase at two times western blotting. **d** All the character of each enzyme and protein complex. The phosphoglycerate mutase–enolase–pyruvate kinase complex has lower $K_M$ and 19 times efficiency compared with free enzyme. **e** Schematic representation of the isotope dilution experiment to assess the channeling of 2PGA and PEP. [13]C-labeled glucose was incubated to yeast HXK, PGI, PFK, ALD, TPI, GAPDH, GAPDH, and PGK with ATP and $NAD_+$ for 2h, and then fed to isolate *Arabidopsis* mitochondria and the label accumulation in pyruvate was monitored. Non-labeled 2PGA and PEP were added into the medium once the fractional enrichment of [13]C label in pyruvate had reached steady state. In the case of channeling, the addition of non-labeled intermediates does not affect the subsequent labeling of pyruvate. **f** The result of isotope dilution experiments for 2PGA. The time-course plot shows the ratio in fractional [13]C enrichment in pyruvate compared with the unlabeled pyruvate at different time points. The metabolite is considered to be channeled when the confidence interval line is ~1. **g** The result of isotope dilution experiments for PEP. The metabolite is considered not to be channeled when the confidence interval line is <1.

increasingly being provided by the analysis of the structures of the constituent proteins of the pathway[4,41–43]. Here, we used a combination of molecular, cell biological, and biochemical evidence to examine the assembly of plant glycolytic enzyme assemblies. In doing so, we were able to demonstrate the presence of a

phosphoglycerate mutase–enolase metabolon as a constituent of a larger complex and to demonstrate that this complex plays a critical role in the colocalization of mitochondria and chloroplasts.

Two of the best-characterized enzyme assemblies, which are conserved across kingdoms, are those formed by constituents of the

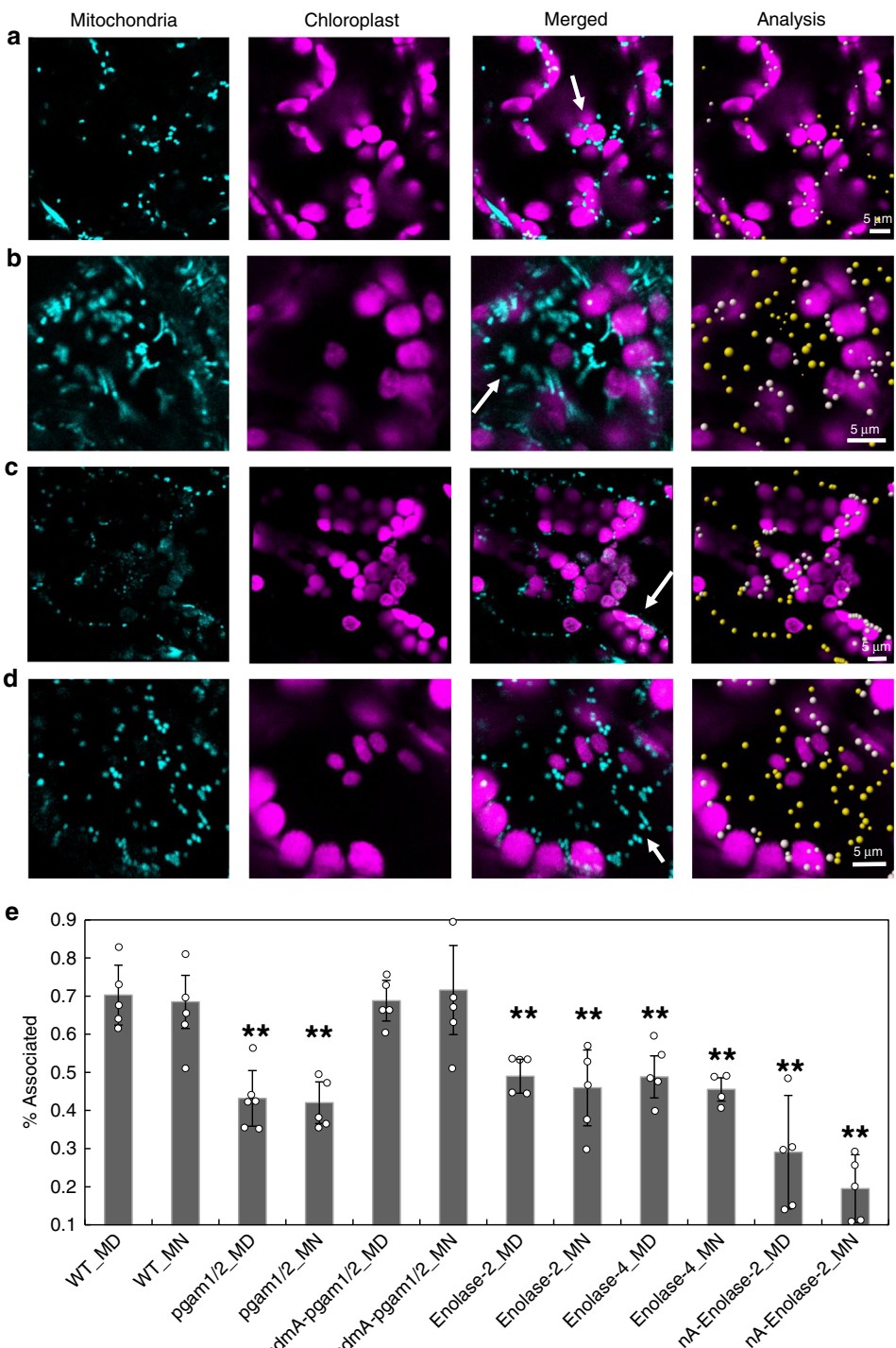

**Fig. 5 The association of mitochondria and chloroplast.** The mitochondria associated at middle of the day (MD) in wild type (**a**); pgam1/2 (**b**), sdmA-pgam1/2 (**c**), the double mutant complemented by native promoter with non-functional phosphoglycerate mutase); enolase-2 (**d**). The cyan fluorescence is the mitochondria. Purple is the autofluorescence. Yellow represents the unassociated mitochondria and white represents associated mitochondria, respectively. **e** the difference of mitochondria associated between WT and mutants. Enolase-2 and enolase-4 are two mutants of enolase. nA-enolase-2 is the enolase-2 complemented by native promoter with nuclear target enolase. One-way ANOVA analysis by WT_MD as control (*$P < 0.05$, **$P < 0.01$, SD). MN middle of the night, pgam1/2 double mutant of phosphoglycerate mutase 1 and phosphoglycerate mutase 2, sdmA-pgam1/2 the double mutant complemented by native promoter with non-functional phosphoglycerate mutase), nA nuclear targeted. Note: scale bar is different in each figure. White arrow is used to highlight the most important points within each panel.

TCA cycle and glycolysis. Strong evidence for metabolite channeling has been provided for the TCA cycle in several organisms[3,14,43]. Proof of the occurrence of substrate channeling in the glycolytic pathway is, however, confined to studies of plant glycolysis. These revealed the association of the entire pathway to the outer mitochondrial membrane of the mitochondria[12], as was also observed in yeast[5] and blood cells[6], and metabolite channeling in the upper half of glycolysis, that is, in the reactions linking glucose 6-phosphate to

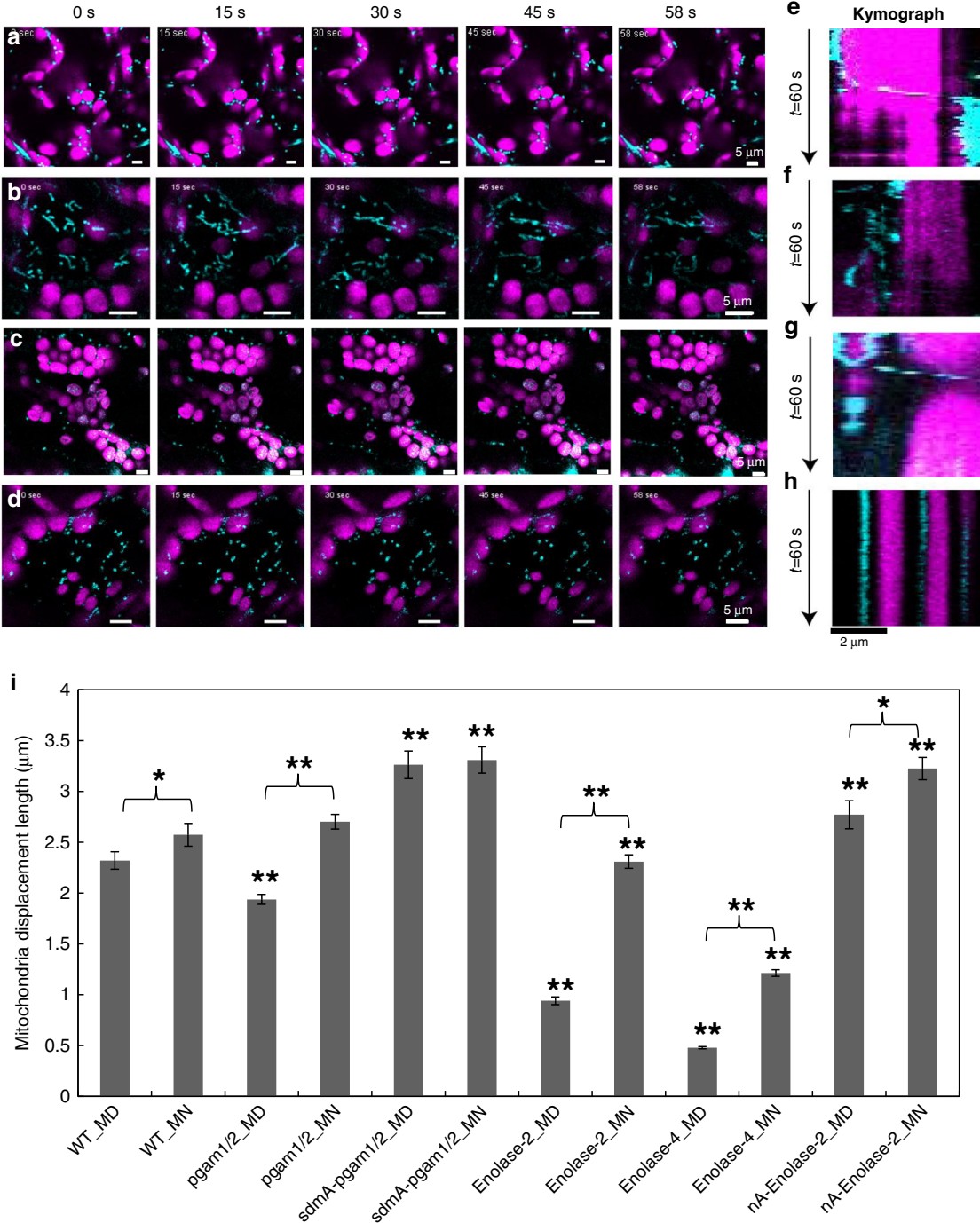

**Fig. 6 The analysis of mitochondrial movement in the plant mature leaves.** Mitochondrial movement analysis 1 min at middle of the day (MD) in wild type (**a**), pgam1/2 (**b**), sdmA-pgam1/2 (**c**), and enolase-2 (**d**). The mitochondria of WT and sdmA-pgam1/2 mostly move around the chloroplast, while the mitochondria of mutants randomly and disorderly moved in the cell. Kymograph of the mitochondrial movement at MD in wild type (**e**), pgam1/2 (**f**), sdmA-pgam1/2 (**g**), and enolase-2 (**h**). The mitochondria attached to the chloroplast at 10–15 s and move to other place both in WT and sdmA-pgam1/2, while there is not attachment between mitochondria and chloroplast in other mutant. **i** Displacement length of all the lines. Displacement is the mitochondria move distance at the 1 min detected time. One-way ANOVA analysis by WT_MD as control for samples of MD or WT_MN as control samples of MN (*$P < 0.05$, **$P < 0.01$, SEM) and mutant's MD was compared with MN. MN middle of the night, pgam1/2 double mutant of phosphoglycerate mutase 1 and phosphoglycerate mutase 2, sdmA-pgam1/2 the double mutant complemented by native promoter with non-functional phosphoglycerate mutase), nA nuclear targeted. Note: scale bar is different for each figure.

glyceraldehyde phosphate[7]. In the current study, we identified a second metabolon channeling 2-phosphoglycerate and demonstrate that this renders the lower part of glycolysis, which is already highly efficient[44], even more so. Our study revealed within-glycolysis associations similar to those that have previously been identified in yeast[5] and human[6,45], as well as interaction with cytoskeletal elements similar to those described in yeast, muscles, and *Arabidopsis*[5,45,46]. Here, we employed the following quantitative methods: AP, BiFC, split-LUC, Co-IP, and FLIM-FRET in order to reliably capture relatively weak interactions since it can be anticipated that

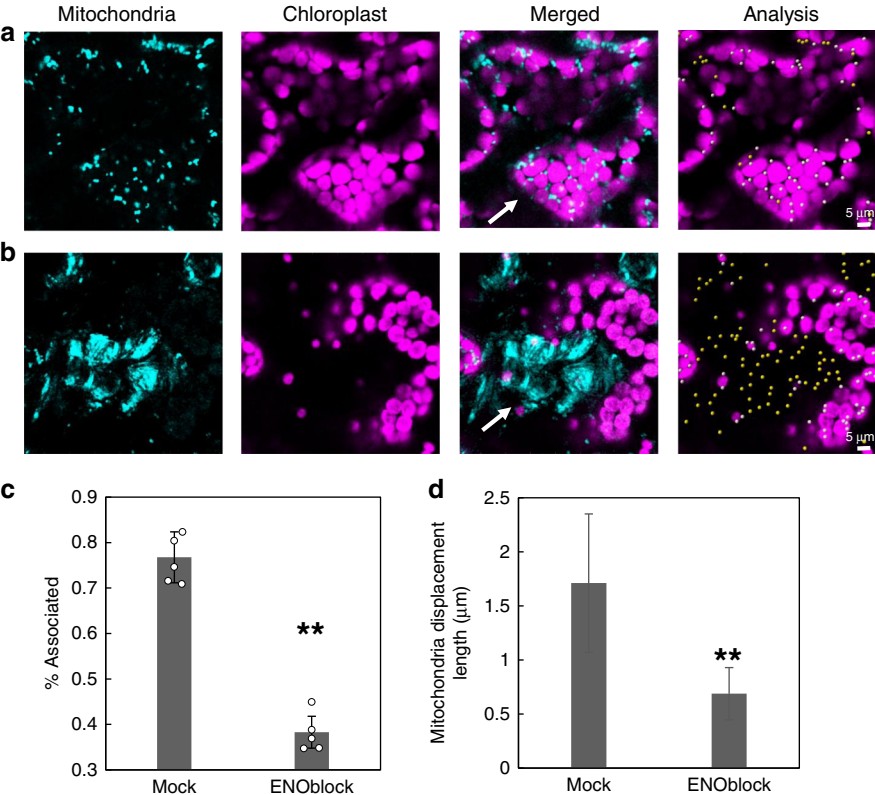

**Fig. 7 The association of mitochondria and chloroplast was significantly decreased by the enolase inhibitor. a** The mitochondria associated at wild type treated with mock ~40 min. **b** The mitochondria associated at wild type treated with enolase inhibitor (ENOblock hydrochloride) ~40 min. **c** The difference of mitochondria associated after treating with mock and ENOblock. One-way ANOVA analysis (**$P < 0.01$, SD). **d** Mitochondrial movement analysis after treating with mock and ENOblock. One-way ANOVA analysis ($P < 0.01$, SD). Note: scale bar is different for each figure. White arrow is used to highlight the most important points within each panel.

employing multiple techniques will have a higher chance of detecting interactions that are affected by protein microenvironments. This enabled the reliable assessment of binary protein–protein interactions of one to dozens of protein pairs and thus facilitated a comprehensive analysis of all possible interactions between proteins constituting and associated with plant glycolysis. While not all the interactions of the glycolytic complex and transporters were detected by all of the methods, a lack of consensus between different methods for identifying protein–protein interactions has previously been documented and likely reflects the high degree of false negatives obtained when basing conclusions on a single methodology[47]. That said, several interesting interactions were confirmed by multiple methods, including not only the phosphoglycerate mutase–enolase–pyruvate kinase interaction and the previously characterized PFK–ALD–triose phosphate isomerase–glyceraldehyde phosphate dehydrogenase complex[7], but also the interactions with actin and tubulin, as well as with the VDAC proteins (Fig. 1a).

Returning to the metabolite channeling occurring between phosphoglycerate mutase and enolase, our studies provide two complementary lines of evidence for this. First, kinetic characterization of the three enzyme complexes resident in mitochondrial preparations revealed that, as would be anticipated from theoretical considerations[48], it was considerably more efficient than the free enzymes of the extra-plastidial preparation (exhibiting approximately five times higher substrate affinity and 19 times higher efficiency), as well as preventing the use of substrate by competing reactions such as PEPC (phosphoenolpyruvate carboxylase) and amino acid biosynthesis. Second and more compelling, isotope dilution experiments wherein $^{13}$C-labeled

3PGA was provided to isolate mitochondria, and the recovery of the label was monitored prior to and following the provision of unlabeled 2PGA or PEP. These experiments suggest that 2PGA is entirely channeled, but that PEP is unchannelled. While four possible rationales are given to explain the value of a channel: regulation, cytotoxic metabolites, unstable metabolites, or leakage impacts the cellular operational efficiency. Which one applies to the metabolon here is unclear; it is possible that there is a thermodynamic benefit. The metabolic intermediates are not believed to be toxic or unstable and their leakage impacting the cell is also unlikely. Moreover, looking at the in vivo thermodynamics of these reactions in other systems[49], data are not available to estimate these in plants, suggesting that there is likely no thermodynamic advantage to metabolon formation. Thus, despite careful consideration, the exact purpose of the metabolite channel currently remains opaque and will require further study to clarify. That said, given that the association and disassociation of metabolon appears to depend on the energy needs of the cell[50], future research should also study if the (dis)assembly of the metabolon are dependent on the in vivo energy requirements of the cell. Finally, given that the phosphoglycerate mutase–enolase–pyruvate kinase is evolutionarily well conserved, we used homology modeling to suggest the structure of the complex (Supplementary Fig. 6). Our model reveals that this channel appears to initiate from the scissor of phosphoglycerate mutase 1 via an encapsulated hole formed by enolase. However, it is important to note that considerable further experimental work is required in order to validate this model.

Considerable recent research has focused on the importance of membrane contact points as important organizing features

involved in organellar interactions[17,19,20]. However, despite evidence being presented for both mitochondrial and chloroplastic membrane contract points, the study of mitochondrial–chloroplast interactions have, to date, received relatively little attention in this vein. Having identified that the phosphoglycerate mutase–enolase–pyruvate kinase interacts with VDAC proteins of the outer mitochondrial membrane and the TPT of the chloroplast led us to take cell biological approaches to evaluate if this complex influences mitochondria–chloroplast interactions. Intriguingly, the data presented here revealed that the phosphoglycerate mutase–enolase–pyruvate kinase subcomplex was an essential component of a larger complex that tethers mitochondria to the chloroplast. While in our study the colocalization of the two organelles was invariant between the light and dark in the WT, there was some diel variation in the degree of colocalization in the mutants between these two conditions. The WT observation is in keeping with observations in some plant cell studies[28,51], while others suggest that the degree of association of the organelles varies depending on the presence of light[52]. Irrespective of the fact that different observations have been made in WT plant cells, there was a clear loss of mitochondrial–chloroplast association in both conditions in *phosphoglycerate mutase* and *enolase* knockout mutants. Our data, when taken collectively with those of previous studies, strongly suggest that the VDAC of the outer mitochondrial membrane act as membrane contact sites and also suggest that the TPT of the inner mitochondrial membrane also interacts with the phosphoglycerate mutase–enolase–pyruvate kinase complex. However, it is important to note that the mechanism of interaction between the subcomplex and the TPT is currently unclear and it is likely to be indirect. Indeed, looking at the distance between mitochondria and chloroplasts, it is clear that these proteins alone are too small to bridge the gap observed between the mitochondria and chloroplasts. In addition, as evidenced in the kymograms[32] of Fig. 6e, lacking either phosphoglycerate mutase or enolase compromised the movement of their mitochondria as well as the degree of mitochondria–chloroplast colocalization. Indeed, the presence of dense bridge structures linking the mitochondria and chloroplasts have recently been observed using TEM to study inter-organelle structural associations in nectary parenchyma cells of *Citharexylum myrianthum*[21]. The cellular phenotypes of the *enolase* mutant could not be complemented by ectopic expression of the nuclear-targeted enolase, but that the mitochondrial movement phenotype could be complemented and both the degree of chloroplast–mitochondrial colocalization and the mitochondrial movement phenotype of the *pgam* mutant could be complemented by a catalytically inactive site-directed mutant of phosphoglycerate mutase. These results suggest colocalization of the mitochondria and cytosol and their degree of movement separable phenomena, with neither being dependent per se on the phosphoglycerate mutase–enolase–pyruvate kinase complex. Moreover, the degree of movement was, at least partially, under the influence of the further moonlighting role carried out by enolase in the nucleus.

Despite the fact that the cellular phenotypes could be complemented using the catalytically inactive phosphoglycerate mutase, it is important to note that the morphological phenotypes of the mutant could be complemented, with the exception of with the native enzyme under the control of its own promoter. This finding demonstrates the general importance of these enzymes for normal plant physiology. Here, we have demonstrated that phosphoglycerate mutase and enolase form a metabolon bound to the outside the mitochondrial via association with pyruvate kinase and the VDAC. We have additionally recovered the previously described plant glycolytic metabolon[7], confirmed interactions between glycolytic enzymes and cytoskeletal components[5,46] and provided evidence of a putative association between the phosphoglycerate mutase–enolase–pyruvate kinase complex and the TPT. In this research, the combination of several binary protein–protein interaction approaches with enzyme activity, isotope dilution experiment, mitochondrial movement analysis, and plant complementation analyses clearly demonstrated the existence of a moonlighting role of lower glycolysis that potentially promotes a highly efficient coordination of the major energy systems of the plant cell. Future work should focus on elucidating the amino acid residues, which are integral to the formation of the complex since only once we know this can we develop the tools to readily ascertain the precise cellular and organismal benefits conferred by the dynamic association between the lower half of glycolysis and membrane proteins of both the mitochondrial and chloroplast membrane systems.

## Methods

**Plant materials**. *Arabidopsis thaliana* genotypes Columbia (Col-0) (WT) and *enolase* (enolase-2-2: SALK_021737; enolase-2-4: SALK_077784[29]), *phosphoglycerate mutase* 1 (pgam1: SALK_003321[30]), and *phosphoglycerate mutase 2* (pgam2: SALK_029822[30]) mutants were used in this study. The seeds were plated on Murashige and Skoog medium supplemented with 1% (w/v) sucrose for 10 days, and then the seedlings were transferred to soil and kept under short day conditions (i.e., a 8-h light (22 °C)/16-h dark (18 °C) photoperiod) in a growth chamber at a light intensity of 120–150 µmol/m²/s. Pre-bolting mature rosette leaves of 30-day-old plants were harvested for metabolite measurements at the beginning of the day. The *Arabidopsis* cell suspension culture (PSB-D) was grown in darkness at an average temperature of 24/25 °C using the standard protocol described at TAIR (https://www.arabidopsis.org).

The cytosolically localized glycolytic isoforms of *Arabidopsis* were selected based on reference to the literature[53]. Expression of genes encoding these proteins was evaluated in the AtGenExpress "development" dataset in order to select those that were ubiquitously expressed. In addition, *E. coli*-pgam (uniport: P62707 and strain: K12), *E. coli*-enolase (uniport: P0A6P9 and strain: K12), the nuclear-targeted enolase, nuclear-targeted ipgam1 under the control of its native promoter, and a site-directed-mutant inactive IPGAM1 (H39, S80, K360, and H470 mutated) were also subcloned; thus, the list totaled 48 proteins (Supplementary Data 1). Full-length coding sequences of these proteins were cloned by PCR-based Gateway BP cloning using the pdonr207 vector. The gene-specific primers used did not include a stop codon to ensure C-terminal fusion of tags (Supplementary Data 2). Expression vectors for AP-MS, Co-IP, BiFC, and split-LUC were constructed using the Gateway LR reaction with pK7FWG2, pK7WG2 (VIB)[54], and pDuExAc6/pDuExDc6[55], respectively. The mCitrine and mCherry[56] were subcloned with the related gene into pdonr207 and pdonr L4-L3 by the In-Fusion (Clontech) method, and then subcloned into pK7m34GW2-8m21GW3 (VIB)[54] by means of the LR reaction (Thermo Fisher Scientific) for the FLIM-FRET and co-sublocalization assays, respectively. The *E. coli*-PGAM was subcloned with *Arabidopsis* enolase promoter and terminator by In-Fusion, and then subcloned into PMDC110[57]. Full-length PGAM1, nuclear-targeted PGAM1, and the site-directed PGAM1 all under the control of their native promoter and terminator were also subcloned into PMDC110[57]. The mitochondrially targeted GFP (Mt-rb)[31] was expressed under the control of the *UBIQUITIN10* promoter into pdnor221 by In-Fusion and BP reaction and subcloned into pMDC110 for mitochondrial movement and associated analysis. All of the vectors constructed in this study are listed in Supplementary Data 3. These constructs were transformed into heterozygous mutants of *pgam1/2* and *enolase* for the further analysis.

**Affinity purification-mass spectrometry**. AP-MS was conducted by expressing target proteins fused with a C-terminal GFP tag in the PSB-D *Arabidopsis* cell culture line using the published method[13,23]. Tandem GFP fused with an N-terminal mitochondrial-targeting peptide was used as a negative control. PSB-D cells (Arabidopsis Biological Resource Center, ABRC) were cultured in the dark at 25 °C with shaking at 120 r.p.m. The cell line was cultured on Murashige and Skoog Basal Salts with minimal organics medium supplemented with 50 µg/l kinetin, 0.5 mg /l 1-naphthaleneacetic acid, and 3% sucrose.

Cells were subcultured every week at a 1:10 culture to fresh media ratio. *Agrobacterium tumefaciens* strain GV3101 transformed with an expression vector was grown on a plate for 2 days and then scratched and resuspended into MSMO (Murashige and Skoog medium with minimal organics) medium to gain an OD600 of 1.0. A 3-ml aliquot of 2-day-old PSB-D cell culture was mixed with 200 µl of *A. tumefaciens* suspension and 6 µl of 100 mM acetosyringone and cocultivated for 72 h. Transformed cells were selected in a medium containing 25 µg/l of kanamycin, 500 µg/l carbenicillin, and 500 µg/l vancomycin for three rounds of a 1-week subculture, followed by those with medium containing only kanamycin for

two rounds[13]. Expression and localization of the tagged proteins were evaluated by viewing GFP fluorescence using confocal microscopy. The transformed cells were collected by vacuum filtration at 5 days after subculturing and frozen in liquid nitrogen. After grinding into a fine powder using a ball mill (MM301, Retch, Haan, Germany), proteins were extracted by mixing 2 g of the material with 2 ml of the extraction buffer (25 mM Tris-HCl, pH 7.5, 15 mM MgCl$_2$, 5 mM EGTA, 1 mM dithiothreitol, and 1 mM phenylmethylsulfonyl fluoride). Following removal of cell debris by repeated centrifugation at $22,000 \times g$ at 4 °C for 5 min, the supernatant was mixed with 25 µl of GFP-Trap_A slurry (ChromoTek, Martinsried, Germany) equilibrated with the extraction buffer and incubated for 1 h at 4 °C with rotation. The beads were collected by centrifugation at $3000 \times g$ at 4 °C for 3 min and washed three times each with the extraction buffer containing 0, 250, and 500 mM of NaCl[13,23].

The proteins that remained on the beads were subsequently subjected to proteomics as in-solution digestion by LysC and trypsin and the resulting peptides were purified[58]. Liquid chromatography with tandem mass spectrometry (LC-MS/MS) analysis was performed on a Q Exactive Plus (Thermo Fisher Scientific). Quantitative analysis of MS/MS measurements was performed with the Progenesis IQ software (Nonlinear Dynamics, Newcastle, UK). Proteins were identified from spectra using Mascot (Matrix Science, London, UK). Mascot search parameters were set as follows: TAIR10 protein annotation, requirement for tryptic ends, one missed cleavage allowed, fixed modification: carbamidomethylation (cysteine); variable modification: oxidation (methionine), peptide mass tolerance = ±10 p.p. m., MS/MS tolerance = ±0.6 Da, allowed peptide charges of +2 and +3. A decoy database search was used to limit false discovery rates to 1% on the protein level. Peptide identifications below rank one or with a Mascot ion score <25 were excluded. Mascot results were imported into Progenesis QI, quantitative peak area information was extracted, and the results were exported for data plotting and statistical analysis. These intensities were filtered against the experiment control and normalized using the spectral index in the CRAPome website[59]. Finally, the possible interactions were scored as fold change-A (FC-A) score calculated by the SAINT algorithm[13,23,59,60] (Supplementary Data 4). Our AP was performed at least three independent biological replicates—with FC-A values being calculated for each individual replicate. All of the FC-A scores of the detected peptides are presented in Supplementary Data 4. The MS proteomics data have been deposited to the ProteomeXchange Consortium via the PRIDE[61] partner repository with the dataset identifier PXD020588.

**Bimolecular-fluorescence complementation**. BiFC constructs were expressed in mesophyll protoplasts, which were generated from the leaves of *Arabidopsis* Col-0 by the Tape-Arabidopsis Sandwich method[62]. Briefly, the lower epidermal surface of a leaf was removed by peeling with a strip of tape fixed to it. The mesophyll cells remaining on the tape were incubated in 20 mM 2-(*N*-morpholino) ethanesulfonic acid (MES) buffer (pH 5.7) containing 1% cellulose (Yakult, Tokyo, Japan), 0.25% macerozyme (Yakult), 10 mM CaCl$_2$, 20 mM KCl, 0.1% bovine serum albumin (BSA), and 0.4 M mannitol with gentle agitation for 20–60 min until the protoplasts were released into the solution. The protoplasts were washed twice with W5 solution (2 mM MES, pH 5.7, 154 mM NaCl, 125 mM CaCl$_2$, 5 mM KCl, and 5 mM glucose), incubated on ice for 30 min, centrifuged, and resuspended into MMg solution (4 mM MES, pH 5.7, 15 mM MgCl$_2$, and 0.4 M mannitol). Protoplasts were transfected with plasmids in a U-bottom 96-well plate by incubating for 5 min at room temperature under the presence of 20% (w/v) PEG4000. Following two washings with W5 solution, the protoplasts were incubated in the dark at 25 °C overnight. The protoplasts were incubated with MitoTracker orange CMTMRos (Thermo Fisher Scientific) for mitochondrial staining at 37 °C for 10 min, followed by 26 °C for 20 min.

The constructs were also transformed into Agrobacteria and infiltrated into *Arabidopsis* leaves for BiFC analysis[25]. Briefly, the *Agrobacterium* was streaked onto YEB agar plates containing 0.1 mM acetosyringone and antibiotics. After 2 days, the agrobacteria were put into wash solution (10 mM MgCl$_2$ and 0.1 mM acetosyringone) for checking OD, and then diluted into an infiltration buffer containing ¼ strength MS with 1% sucrose, silwet L-77, and 0.1 mM acetosyringone at OD 0.5. The transformed agrobacteria were infiltrated into 3-week-old *Arabidopsis* leaves, then kept in the dark for 24 h prior to being left in the greenhouse to recover for 2 days[25]. Confocal images were taken using a DM6000B/SP8 confocal laser scanning microscope (Leica Microsystems, Wetzlar, Germany). BiFC fluorescence was imaged with a 488-nm laser excitation and emission fluorescence was captured by 500–520-nm band-pass emission filters, respectively.

**Split-LUC complementation assay**. The plasmids for split-LUC assays were extracted from bacterial cells by alkaline lysis and purified using silicon dioxide (Sigma-Aldrich) slurry[63]. Mesophyll protoplasts were generated from the leaves of *Arabidopsis* Col-0 by the Tape-Arabidopsis Sandwich method as mentioned above[62]. For luminescence detection, 10 µl of 60 µM ViviRen Live Cell substrate (Promega, Madison, WI) was added to each well of a 96-well plate[14,55]. The plate was kept for 4 min at room temperature in the dark before measuring the luminescence using a CLARIOstar microplate reader (BMG LABTECH, Ortenberg, Germany) with 10-s integration periods at emission 480±30 nm. The experiments were repeated four times.

**Co-IP with western blotting**. Two constructs were separately transformed into Agrobacteria and infiltrated into *Arabidopsis* leaves[25]. Plant materials were harvested in liquid N$_2$ and the powder was extracted with protein extraction buffer (25 mM Tris-HCl, pH 7.5, 15 mM MgCl$_2$, 5 mM EGTA, 1 mM dithiothreitol, and 1 mM phenylmethylsulfonyl fluoride) on ice for 5 min and centrifuged twice for 10 min at $12,000 \times g$ at 4 °C. After determining the protein concentration, proteins were subsequently separated on 10% sodium dodecyl sulfate-polyacrylamide gel electrophoresis and transferred to a PVDF membrane (Bio-Rad) for 60 min, at 300 mA in temperature-controlled conditions by transmembrane buffer. The membrane was blocked using incubation with 5% defatted milk for 1 h. The primary antibody (1:5000), dissolved in 5% skimmed milk in TBST (50 mM Tris, 150 mM NaCl, 0.1% Tween-20, pH 7.5), was incubated at 4 °C overnight. After washing three times for 10 min with TBST, the secondary antibody (1:3000) was added in 1% skimmed milk-TBST solution and incubated for 1 h, followed by three additional washing steps of 10 min with TBST. For detection, the two-component reagent Clarity Western ECL Substrate (Thermo Fisher Scientific) was used. The signal was exposed and detected with an X-ray film. The antibodies of enolase (Agrisera) and GFP (Sigma-Aldrich) were used in this study.

**Förster resonance energy transfer-fluorescence lifetime imaging microscopy**. Constructs for FLIM-FRET recordings (Supplementary Data 3) were constitutively expressed in plant cell culture alongside negative control (enolase-mCitrine and mCherry), donors, and acceptors, respectively. Fluorescence decay times of the donor molecules ($\tau_D$) were recorded using FLIM in order to determine the corresponding FRET efficiencies. FLIM was performed by the time-correlated single-photon counting technique using the MicroTime 200 system (PicoQuant, Berlin, Germany) with an instrument response function of 250 ps. Samples were excited at $\lambda_{ex} = 475$ nm and fluorescence was detected at $\lambda_{em} = 536 \pm 20$ nm, minimizing the influence of chlorophyll fluorescence. Image acquisition occurred at 128 pixel × 128 pixel within 232 s controlled by the SymPhoTime 64 software (PicoQuant, vers. 1.6) in order to obtain appropriate photon numbers for reliable fluorescence decay analyses. Data were analyzed using deconvolution fitting model implemented in the software. Here, a biexponential fluorescence decay behavior has been observed for all combinations and for further calculations, an amplitude-weighted average decay time was determined. Regions of interest (ROIs) from 10 to 23 different samples of each combination of target and control proteins were analyzed (except for WT $N = 5$ and only acceptor $N = 5$). Then, average values of all analyzed ROIs were used to calculate a mean FRET efficiency. The FRET efficiency was calculated using the equation (Supplementary Data 5): $E_{FRET} = 1 - (\tau_{DA}/\tau_D) \times 100$ [%]; with $t_{DA}$ the decay time of the donor in the presence of the acceptor, and $t_D$ the decay time of the donor in the absence of acceptor. The minimum and maximum $E_{FRET}$ values obtained using this experimental setup were defined by recording the FRET efficiencies between free soluble mCitrine and mCherry (0%), as well as between a fusion construct of mCitrine and mCherry (20%) (Supplementary Data 5), corresponding to intermolecular distances >15 nm and <6 nm, respectively, calculated as the center-to-center distances between the fluorescent proteins.

**Co-sublocalization**. Constructs for FRET-FLIM recordings (Supplementary Data 3) were transient expressed in the *Arabidopsis* leaves by agroinfiltration[25] for the protein co-sublocalization analysis. Enolase-mCitrineNE/PGAM1-mCitrineCE and PYK4-mCherry constructs were coinfiltrated into *Arabidopsis* leaves in order to assess if the three proteins co-sublocalized. Confocal images were taken using a DM6000B/SP8 confocal laser scanning microscope.

**Mitochondrial purification and enzyme activity assays**. Phosphoglycerate mutase assays were performed by coupling the formation of lactate from 3-phosphoglycerate with enolase (Sigma-Aldrich), pyruvate kinase (Sigma-Aldrich), and lactate dehydrogenase (Sigma-Aldrich)-catalyzed reactions at room temperature and measured at 340 nm according to the protocol of enzymatic assay of phosphoglycerate mutase (Sigma EC 5.4.2.1)[30,64]. The assay consisted of adding 5 µl of desalted extract to the phosphoglycerate mutase assay buffer (MgCl$_2$ 10 mM, HEPES/KOH 0.05 M, pH 7.5, EDTA 2 mM, Triton X-100 0.05%, NADH 1 mM, ADP 1 mM, 1 U enolase (Sigma), 1 U pyruvate kinase (Sigma), 1 U lactate dehydrogenase (Sigma)). The reaction was started by the addition of 3PGA to a final concentration of 3 mM. Enolase assays were performed by coupling the formation of lactate from 2-phosphoglycerate with the pyruvate kinase (Sigma-Aldrich), and lactate dehydrogenase (Sigma-Aldrich)-catalyzed reactions at room temperature and measuring absorbance at 340 nm[44]. The assay consisted of adding 5 µl of desalted extract into enolase assay buffer (MgCl$_2$ 10 mM, HEPES/KOH 0.05 M, pH 7.5, EDTA 2 mM, Triton X-100 0.05%, NADH 1 mM, ADP 1 mM, 1 U pyruvate kinase (Sigma-Aldrich), 1 U lactate dehydrogenasae (Sigma-Aldrich)). The reaction was started by the addition of 2PGA to a final concentration of 1.5 mM. Pyruvate kinase assays were performed by coupling the formation of lactate from phosphoenolpyruvate with the lactate dehydrogenase (Sigma-Aldrich)-catalyzed reactions at room temperature and measuring absorbance at 340 nm. The assay consisted of adding 5 µl of desalted extract into pyruvate kinase assay buffer (MgCl$_2$ 10 mM, HEPES/KOH 0.05 M, pH 7.5, EDTA 2 mM, Triton X-100 0.05%, NADH 1 mM, ADP 1 mM, 1 U lactate dehydrogenase (Sigma-Aldrich). The

reaction was started by the addition of *phosphoenolpyruvate* to a final concentration of 1 mM.

Mitochondria were isolated from the aerial part of 7-week-old *Arabidopsis* plants grown under short-day photoperiod conditions as described previously[65]. Briefly, ~30 g of whole rosette leaves was disrupted with a tissue grinder in 150 ml of cold extraction buffer [0.3 M sucrose, 5 mM tetrasodiumpyrophosphate (10 $H_2O$), 2 mM EDTA, 10 mM $KH_2PO_4$, 1% polyvinylpyrrolidone (PVP-40), 1% BSA, 20 mM ascorbic acid, 5 mM cysteine (pH 7.5)]. The homogenate was filtered twice through four layers of Miracloth (CalBioChem). The material that was retained was recovered and ground in a cold stone mortar with extraction buffer. This step was repeated three times. The preparation was centrifuged first at $1100 \times g$ for 10 min to separate chloroplast (pellet) and mitochondrial (supernatant) fractions. The supernatant containing the mitochondria-enriched fraction was further centrifuged for 10 min at $18,000 \times g$. The pellet was resuspended in a small volume of washing buffer [0.3 M sucrose, 10 mM 3-(*N*-morpholino)propansulfonic acid (MOPS), 1 mM EGTA (pH 7.2)] and homogenized with a Potter–Elvehjem homogenizer. An additional 15 ml of washing buffer was added, and the mixture was centrifuged again at $1100 \times g$ for 10 min. The supernatant was transferred to new tubes (40 ml), making sure that no pellet was transferred. The suspension was then centrifuged at $18,000 \times g$ for 10 min. The obtained pellet was resuspended in 1 ml of washing buffer and loaded on top of a Percoll (GE Healthcare) step gradient. Each gradient consisted of mitochondria gradient buffer [1.5 M sucrose, 50 mM MOPS (pH 7.2)] and a 0–4.4% (v/v) PVP gradient in 28% (v/v) Percoll (1). The gradient loaded with the sample was centrifuged at $40,000 \times g$ for 45 min. The bottom part of the gradient containing the mitochondria was collected, washed three times, divided into aliquots for the enzyme activity and isotope dilution experiment.

The enzyme activity of the protein complex and free enzyme were performed as for the phosphoglycerate mutase assay without the *recombinant* enolase and *recombinant* pyruvate kinase with isolated mitochondria and total extra-plastidial protein extracts. The enzyme activity was measured as detailed above. The reaction was started by the addition of 3PGA to a final concentration of 3 mM. The efficiency of the complex was contrasted with the $K_M$ of each individual enzyme for comparative purposes. The 3PGA utilization model assumed that 20% of enzymes formed a metabolon at the outer membrane of the mitochondria. The substrate utilization was calculated using the equation: products = $V_{max} \times [s]/(K_M + [s]) \times E$, with $E$ representing the ratio of the enzyme in the isolated mitochondria to that in the total extra-plastidial protein extracts.

**Isotope dilution experiments**. $^{13}C$-labeled glucose (100 mM) was incubated with recombinant glycolysis enzymes (hexokinase, phosphoglucose isomerase, PFK, ALD, tripsephosphate isomerase, glyceraldehyde phosphate dehydrogenase, phosphoglycerate kinase (Sigma-Aldrich)) and 200 mM ATP and 100 mM $NAD^+$ at the enzyme activity buffer ($MgCl_2$ 10 mM, HEPES/KOH 0.05 M, pH 7.5, EDTA 2 mM, Triton X-100 0.05%) for 2 h prior to the isotope dilution experiment to produced labeled $^{13}C$ 3PGA ($54 \pm 18$ mM). Mitochondria were isolated as mentioned above. The isolated mitochondria were incubated in the enzyme activity buffer for 1 h until the label accumulation in the product reached an isotopic steady state (i.e., the fractional enrichment of label was constant with time). Then, a 10 mM unlabeled intermediate—either 2PGA or PEP—was added and the "dilution effect" on the labeling in the product was monitored over time. The reactions were stopped at different time points in liquid nitrogen and followed by the gas chromatography-MS (GC-MS) strand extraction and analysis. The ration of $^{13}C$-labeled pyruvate and unlabeled pyruvate was calculated to analyze the channeling status.

**Mitochondrial movement analysis**. The mitochondrially targeted GFP[31] under the control of the *UBIQUITIN10* promoter was transformed into the heterozygous *pgam1/2*, *enolase-2*, and *enolase-4* mutants. The site-directed mutant of PGAM1 was transformed into heterozygotes of *pgam1/2*, and then crossed with pgam1/2 of mitochondrially targeted GFP for the mitochondrial movement analysis. The nuclear localized enolase was transformed into heterozygotes of *enolase-2* and then crossed with enolase-2 of mitochondrially targeted GFP for the mitochondrial movement analysis. The below phase of the T3 plants leaves are used for confocal imaging. Ten days of seedlings of WT-GFP or actin marker lines were treated with 200 μM ENOblock (AP-III-a4 hydrochloride, Cat. no. HY-15858A) for 40 min for the mitochondrial movement or cytoskeleton analysis. Time-lapse imaging was performed at a time interval of every 0.86 s for a period of 1 min with a ×63 water objective using the 488-nm Argon laser for exciting GFP. GFP emission was collected between 498 and 550 nm and autofluorescence from chloroplast was detected between 600 and 700 nm. Image analysis was performed using IMARIS ®BITPLANE. To estimate the association between chloroplast and mitochondria, spot detection algorithm was used to detect mitochondria with a spot diameter ranging between 0.5 and 0.75 μm after background subtraction. A quality threshold >38 was set based on manual inspection that showed optimal mitochondrial detection. The chloroplast was detected using the surface detection function with a manual threshold set between 100 and 255, which covered the surface volume of the chloroplast. In addition to this, a quality filter was used to eliminate incorrect surface detection. Upon detection of mitochondria as spots and chloroplast as surfaces, a built-in MATLAB algorithm was used to estimate mitochondria (spots) in association with chloroplast (surfaces) based on a threshold value of 1 that was estimated after thorough visual analysis of colocalization of the different threshold

values. Each genotype and treatment contained four to six explants with >1000 mitochondria subjected to analysis. For measurement of mitochondrial movement, mitochondria was detected as spots using similar threshold values as above and tracked in the subsequent time points using autoregressive motion algorithm with a gap distance of three frames using the quality threshold function in IMARIS. It is important to note that in all cases thorough manual checks were performed to ensure proper detection and tracking based on the applied threshold and filters. A minimum of 300 mitochondria were tracked for each genotype and treatment.

**Gas chromatography-mass spectrometry**. Metabolite profiling of *Arabidopsis* leaves was carried out by GC-MS (ChromaTOF software, Pegasus driver 1.61; LECO) as described previously[66]. The chromatograms and mass spectra were evaluated using the TagFinder software[67]. Metabolite identification was manually checked by the mass spectral and retention index collection of the Golm Metabolome Database[68]. Peak heights of the mass fragments were normalized on the basis of the fresh weight of the sample and the added amount of an internal standard (ribitol). Relative metabolite levels were obtained as the ratio between the lines and the mean value of the respective WT. Statistical differences between groups were analyzed by Student's *t* tests.

**Protein structure**. Structures of the TCA cycle and glycolysis enzyme were predicted by package of Iterative Threading ASSEmbly Refinement (I-TASSER)[69], which is a hierarchical approach to protein structure and function prediction. Structural templates were first identified from the PDB by multiple threading approach LOMETS[70], which generated 3D models by collecting high-scoring target-to-template alignments from locally installed threading programs. The full-length atomic models were then constructed by iterative template fragment assembly simulations. The high intensities and cover structure could be selected as the model of target protein. The complex structure were rebuilt by PYMOL[71] and ClusPro package[72], which is a web server for rigid-body protein–protein docking that combines computational and experimental information.

**Statistical analyses**. All the data are analyzed by the one-way analysis of variance feature of Sigma plot (Systat Software Inc.). The term "significant" is used in the text only when the change in question has been confirmed to be significant ($P < 0.05$) with this test. Single asterisk represents $P < 0.05$ and double asterisks represent $P < 0.01$. Most of the error bar used SD (standard deviation), except that the mitochondria displacement analysis and the FLIM-FRET analysis used SEM (standard error of the mean). The western blotting results of the mitochondria and cytosol were analyzed by ImageJ to compare with the protein expression levels.

**Reporting summary**. Further information on research design is available in the Nature Research Reporting Summary linked to this article.

## Data availability

The AP-MS data are available at PRIDE (https://www.ebi.ac.uk/pride/) with the dataset identifier PXD020588. Source data are provided with this paper.

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

## Acknowledgements

This work was supported by funding from the European Union's Horizon 2020 research and innovation program, project PlantaSYST (SGA-CSA No. 739582 under FPA No. 664620, Y.Z and A.R.F), the Max Planck Society (Y.Z., A.G., and A.R.F.), German Academic Exchange Service (DAAD; Y.Z.), International Max Planck Research Schools Ph.D. program (IMPRS; Y.Z.). We thank Prof. Sarah M. Assmann (Penn State University Department of Biology), who provided the pgam1 (SALK_003321 and SALK_029822), pgam2 (SALK_016231 and SALK_002280), and pgam1/2 double mutants. We would like to thank Prof. Brigitte Poppenberger (Technische Universität München for providing the enolase mutants (SALK_021737 (enolase-2) and SAIL_208_B09 (enolase-4)).

## Author contributions

Y.Z. and A.R.F. designed the experiments. Y.Z. performed cDNA cloning, vector construction, developed and conducted protein–protein interaction assays, enzyme activity, and performed T-DNA identification, complementation, and GC-MS. S.M.-L.K. purified the mitochondria. C.H. performed the FLIM-FRET analysis Y.Z. and A.S. performed and analyzed the mitochondrial movement. Y.Z., A.S., and K.S. performed ENOblock experiment. Y.Z., C.S., and A.G. performed the LC-MS/MS analysis. L.S. performed critical reading and contributed to manuscript writing. Y.Z. and A.R.F. wrote the manuscript.

## Funding

## Competing interests

The authors declare no competing interests.
