## [Peer Review File · Nature Communications]

Reviewers' comments:

Reviewer #1 (Remarks to the Author):

The authors extensively pursue support for the metabolon formation on the exterior of the mitochondria and serendipitously discover a bridge between the mitochondria and chloroplast based on PGAM-ENOLASE complex interactions with organelles. At least four approaches are used to confirm the channeling mechanism for 3PGA to PEP. Possibly most conclusive would be the isotope dilution studies, however they are partly eroded because of the in vitro nature of the experiments which can affect spatial attributes and create artefacts and may be at the heart of a question and possible inconsistency with thermodynamics.

Points for further consideration:

It is curious that the 3PGA is channeled to PEP not all the way to PYR on the surface of the mitochondria (as is hypothesized in prior literature). The delta G for reactions in glycolysis are well known and descriptions indicate that there is a large negative delta G in the PEP to pyruvate step that 'pulls' glycolysis forward. It is hard for me to reconcile the channel to PEP with thermodynamics because the channel and the cytosolic path would have a common product of PEP, the intermediate that is pulled to pyruvate and driving flux through glycolysis. It would seem like the channel offers no benefit over the bulk pool and I wonder if this can somehow be an artefact of the in vitro nature of some experiments, concentrations and organelle membrane preparations, or what the proper interpretation might be? Also, how would channeling 2PGA away from the cytosol be of value?

Apart from this, pertaining to imaging, should there be a negative control or two to preclude that the enzymes do not just have general affinity for membranes? Basically, what is to say that the binding is not non-specific? Did you visualize the vacuole, or cell wall, or peroxisome or others? This should also be considered for the proposed linkage between organelles by the PGAM-enzyme complex.

Minor:

Use arrows or other markings to highlight the most important points within panels of figures that serve as examples to make your case and describe these in text.

The non-catalytic PGAM complementation experiment is nice. It would be interesting (as the authors mention) to more thoroughly investigate the portion of sequence responsible for the interaction, but beyond the scope of these studies.

Reviewer #2 (Remarks to the Author):

This manuscript has two findings.

The first is a comprehensive analyses of the interaction of glycolytic enzymes and the extension of metabolite channelling. For this finding the data and interpretation are justified and build on previous findings in plants and other system. This is somewhat novel and significant.

The second finding is far more novel and significant – that the glycolytic enzymes build a “bridge” between mitochondria and chloroplasts. This is very interesting as it may reveal a new communication/interaction pathway between organelles. As such while I am excited by this concept, the evidence is very indirect, essentially Figure 3, 5 and to a lesser extend Figure 6.

Figure – definitely co-localisation but not at a resolution that could be considered to be just

Figure 5 – Given the different number/size of mitochondria in each image and association I am not convinced by this. I realise that there would be a number of images etc that would be analysed but I would say B has at least the same amount of contact or in fact more with chloroplasts. While of course statistical analyses needs to be carried out – it must make sense when you look at the image.

The concerns are the mutants are retarded in growth compared to the wild type – Supplemental Figure S7 and Figure S8. This severe growth phenotype will have significant impact on many aspect in the cell – does it affect mitochondrial mass, morphology, number (I realise all are linked), and for chloroplast are they affected in mass, size etc. Does it affect cell size, cell division etc.

Figure 6 – Agreed that mobility is affected, this relates to Figure 5 and 6 it appears that the morphology of the phosphoglycerate mutant (but not the enolase mutant) is greatly affected – given the proteins involved in mitochondrial fusion and fission bind to mitochondria (and chloroplast, and peroxisomes), have the mutants impacted this association and the differences observed are secondary in nature. Also how is binding of mitochondria to the components of the cytoskeleton affected.

Thus, overall what cannot be ruled out, and is a possibility is that the cell biology of these mutants is radically altered, as would be expected from such a fundamental pathways being disrupted.

Independent lines of evidence are needed – as the authors obtained for the interaction data.

To support the claims that the author state there needs to be independent evidence that would rule out or at least make it much less likely. How the authors would like to do this is up to them but what would convince me is to have a mutant that has lost activity (point mutation in the active site) and carry out this study. That way the enzyme is there but the activity isn't. Or the converse have the activity without the domain that interacts with the other enzymes. Other approaches is detailed EM work with immunolabelling. Biochemical approaches include cross-linking to pull down these structures intact.

Reviewer #3 (Remarks to the Author):

This manuscript provides compelling evidence for both metabolite channelling between the glycolytic enzymes PGM and enolase (by using stable isotope labelling to demonstrate that the substrates have decreased interaction with the bulk phase). Critically, they also show that these enzymes are also involved in a physical interaction between chloroplasts and mitochondria. Together, these observations provide a very important advance in knowledge about the spatial organisation of metabolism. I do not have any major criticisms of the methods or interpretation of the results.

I have a few minor comments that the authors may wish to address.

Lines 64-65. It is not clear what point is being made and it is not immediately obvious from the references. I am wondering if this is a reference to the “chemotactic” movement of enzymes along substrate gradients- which has been proposed but could be an artifact of the measurement method (discussed in Smirnov, 2019).

Line 218. “hypothesis” might be better than “theory” in the context.

Lines 343-345. I am not sure that the resolution of microscopy used here can provide information on the size of protein complexes that could bridge or attach the mitochondria and chloroplasts.

Figure 5 and 6 legends. It would help to identify all the abbreviations of enzyme constructs used in the figures.

Reviewer #1 (Remarks to the Author):

The authors extensively pursue support for the metabolon formation on the exterior of the mitochondria and serendipitously discover a bridge between the mitochondria and chloroplast based on PGAM-ENOLASE complex interactions with organelles. At least four approaches are used to confirm the channeling mechanism for 3PGA to PEP. Possibly most conclusive would be the isotope dilution studies, however they are partly eroded because of the in vitro nature of the experiments which can affect spatial attributes and create artefacts and may be at the heart of a question and possible inconsistency with thermodynamics.

Response:

Response: All substrate channeling experiments reported to date have been performed in vitro, either in a cell extract or using isolated organelles (Wheeldon et al., 2016). The only exception we are aware of is the work by Daniel Kohl's group (Shearer et al., 2005) in which a transport mutant of *E.coli* was used that constitutively takes up C6 sugar phosphates to enable uptake of labelled and unlabeled substrates. We are not aware of a similar approach having been successfully used in plants and moreover the experiments are difficult to interpret due to rapid loss of label signatures through the wider metabolic network. Recently published methods reliant on mass spectral imaging (Pareek et al., 2020) are only capable of demonstrating local enrichment in metabolite levels and are not a direct, quantitative measure of substrate channeling. Indeed, to our knowledge, all the carefully characterized plant metabolons have relied on the in vitro isotope dilution approach including the pathways of glycolysis (Graham et al., 2007), the TCA cycle (Zhang et al., 2017), the upper pathway of phenylpropanoid biosynthesis (Achnine et al., 2004) and the cyanogenic glucoside biosynthetic pathway (Laursen et al., 2016). Nevertheless, we accept the criticism that the spatial arrangements of the enzymes and their properties may not have been completely preserved in vitro and we have slightly toned down our claims and acknowledge that future research should study if the (dis)assembly of the metabolon is dependent on the in vivo energy requirements of the cell (Line 322-325).

Points for further consideration:

It is curious that the 3PGA is channeled to PEP not all the way to PYR on the surface of the mitochondria (as is hypothesized in prior literature). The delta G for reactions in glycolysis are well known and descriptions indicate that there is a large negative delta G in the PEP to pyruvate step that 'pulls' glycolysis forward. It is hard for me to reconcile the channel to PEP with thermodynamics because the channel and the cytosolic path would have a common product of PEP, the intermediate that is pulled to pyruvate and driving flux through glycolysis. It would seem like the channel offers no benefit over the bulk pool and I wonder if this can somehow be an artefact of the in vitro nature of some experiments, concentrations and organelle membrane preparations, or what the proper interpretation might be?

Response: The reviewer brings up an interesting, yet complex, aspect. It may well be that the metabolite channel serves a purpose other than a thermodynamic one. Indeed there are several proposed benefits of metabolite channels that are independent of thermodynamics including (i) metabolic regulation - which the reviewer points out is unlikely the case in the scenario we present here, (ii) protecting the cell from cytotoxic compounds, (iii) when compounds are unstable it is advantageous that they are transferred quickly from one active site to another, (iv) potential leakage of metabolites through membranes can render the cell energy and carbon inefficient.

That said metabolite channels have previously been argued to be a route by which to overcome thermodynamically unfavorable reactions (Bar-Even et al., 2012; Noor et al., 2014). Looking at Table 1 below this would be particularly important for reactions 8 and 9 of glycolysis but not reaction 10 which is highly favorable. This may explain the channeling between reactions 8 and 9 but not all the way to pyruvate.

A further complexity of this discussion is that standard Gibbs free energy changes provide some indication of reaction thermodynamics, but to assess this properly one has to look at the thermodynamics in vivo. A range of studies have achieved this in microbes and even in a mammalian cell line (Jacobson et al., 2020; Park et al., 2016; Park et al., 2019). Their findings indicate, perhaps unsurprisingly, that the in vivo ΔG 's are much less unfavorable than indicated by standard Gibbs free energy changes. However, the necessary data to calculate in vivo ΔG 's is not yet available in plants

rendering it difficult for us to comment directly on this aspect. Of course one of the reasons underlying the difference in standard and in vivo ΔG could be the presence of 3-PGA to PEP channeling.

Table 1 Change in free energy for each step of glycolysis (Garrett, 2005)

Step	Reaction	ΔG° / (kJ/mol)	ΔG / (kJ/mol)
8	3-Phosphoglycerate ³⁻ → 2-Phosphoglycerate ³⁻	4.4	0.83
9	2-Phosphoglycerate ³⁻ → Phosphoenolpyruvate ³⁻ + H ₂ O	1.8	1.1
10	Phosphoenolpyruvate ³⁻ + ADP ³⁻ + H ⁺ → Pyruvate ⁻ + ATP ⁴⁻	-31.7	-23

Also, how would channeling 2PGA away from the cytosol be of value?

Response:

Following the reviewers train of thought the PEP to pyruvate reaction is highly thermodynamically favorable and thus this reaction may not need be channeled in order to ensure pyruvate availability for the mitochondria but as the table above shows the two reactions proceeding this one have unfavorable thermodynamics and thus their channeling may prove to be a boon. Given the complexities of these arguments and the fact that (as we state above) we have no evidence that the rationale for the formation of these metabolons is a thermodynamic one we did not include these discussions in the revised manuscript. We would, however, be happy to do so if the reviewer felt them crucial. An example of another potential explanation is that plants have a much higher demand for biosynthesis than other species. Therefore, it is conceivable that not only do they attempt to optimize pyruvate production but also that of PEP to support anapleurotic fluxes. Given that this is speculative we did not comment on it in the manuscript but would be happy to do so if required.

Apart from this, pertaining to imaging, should there be a negative control or two to preclude that the enzymes do not just have general affinity for membranes? Basically, what is to say that the binding is not non-specific?

Response:

In the SUBa4, these glycolytic enzymes are only subcellularly localized in the cytosol, at the plasma membrane and outer membrane of mitochondria (Figure 1B). They have not been reported to associate with the chloroplast or peroxisomes in proteomic experiments focused on these compartments (Hooper et al., 2017). We further co-expressed markers of the mitochondria and the peroxisome or plasma membranes. Based on images of dual localization of mitochondria and peroxisome markers, we do not see significant overlap between the two organelles as compared to the association between mitochondria with chloroplast (Figure S6). Moreover, we used the enolase inhibitor (ENOblock) to treat mitochondrially targeted GFP lines, which is the mitochondria target GFP under the control of the UBIQUITIN10 promoter in wild type col-0 background. Following treatment with the enolase inhibitor associations between mitochondria and chloroplast are significantly decreased. (line 244-251)

Did you visualize the vacuole, or cell wall, or peroxisome or others? This should also be considered for the proposed linkage between organelles by the PGAM-enzyme complex.

Response:

The subcellular location of mitochondria is around the chloroplast and the inner cell membrane (Oikawa et al., 2015). Moreover, both peroxisome and mitochondria could associate to chloroplast (Oikawa et al., 2015). In order to address the reviewers' question as to whether this reflected a general association with membranes we transiently expressed markers of mitochondria and peroxisome but did not visualize any interaction since the peroxisome is also dynamically moving in the cytosol (Figure S6A). In our studies, most mitochondria are associated to the chloroplast in plant cells (Figure S6B). This finding is similar to several published plant cell biology papers (Islam & Takagi, 2010; Oikawa et al., 2015; Ruberti et al., 2014; Sharma et al., 2018). Indeed, here we solely detected an association between mitochondria and chloroplast. However, from our results we cannot formally exclude the possibility of a very weak effect interaction between plasma membrane and mitochondria. Irrespective of whether there is an interaction or not the fact remains that this interaction may be not directly affected by the PGAM-enzyme complex. Comments to this effect have been added to the revised manuscript. (line 251-254)

Minor:

Use arrows or other markings to highlight the most important points within panels of figures that serve as examples to make your case and describe these in text.

Response: We added in figure 3 and 5 a, b, c and d, Figure S3, a, b, c, d and e.

The non-catalytic PGAM complementation experiment is nice. It would be interesting (as the authors mention) to more thoroughly investigate the portion of sequence responsible for the interaction, but beyond the scope of these studies.

Response:

It's hard to find out the portion of sequencing as the transient association of the metabolon as the exactly protein structures of the plant PGAM, enolase and pyruvate kinase of Arabidopsis are still unclear. As the reviewer stated this is however beyond the scope of the current study and further research is still need to continue address these issues.

Reviewer #2 (Remarks to the Author):

This manuscript has two findings.

The first is a comprehensive analyses of the interaction of glycolytic enzymes and the extension of metabolite channelling. For this finding the data and interpretation are justified and build on previous findings in plants and other system. This is somewhat novel and significant.

The second finding is far more novel and significant – that the glycolytic enzymes build a “bridge” between mitochondria and chloroplasts. This is very interesting as it may reveal a new communication/interaction pathway between organelles. As such while I am excited by this concept, the evidence is very indirect, essentially Figure 3, 5 and to a lesser extend Figure 6.

Response:

We understand the reviewer's concern. However, the dynamic movement of both organelles and the transient association of the metabolon, render it exceptionally difficult to provide directly evidence that mitochondrial anchoring to chloroplasts is only dependent on the glycolytic metabolon. However, we believe that the experiments presented indicate that the association of PGAM-enolase-pyruvate kinase metabolon acts as a bridge between the mitochondria and chloroplast. To further test this we used the enolase inhibitor (ENOblock) to treat lines expressing a mitochondrially targeted

GFP. Following these experiments we can conclude that both the association and the movement between mitochondria and chloroplast are significantly decreased following a deficiency of cytosolic enolase and a reduced association with PGAM and PK,. (line 244-251)

Figure – definitely co-localisation but not at a resolution that could be considered to be just

Response: We are not entirely sure what figure the reviewer actually refers to here. We believe that the resolution is correct for every interpretation we made. However, if the reviewer specifically refers to the colocalization between chloroplast and mitochondria, this is more than sufficient. Kymographs in figure 6 E to H and supplemental figure S4 show sustained association between the two over time and that they are not coming into contact randomly. Showing colocalization events in fixed samples will not capture this level of behavior as in these we cannot rule out random associations. Furthermore, the co-localization of mitochondria and chloroplast has been reported by several other papers (Islam & Takagi, 2010; Oikawa et al., 2015; Ruberti et al., 2014; Sharma et al., 2018) .

Figure 5 – Given the different number/size of mitochondria in each image and association I am not convinced by this. I realise that there would be a number of images etc that would be analysed but I would say B has at least the same amount of contact or in fact more with chloroplasts. While of course statistical analyses needs to be carried out – it must make sense when you look at the image.

Response:

We realized that the scale-bar of all the figures are different and noted at the manuscript (Figure 5, 6, 7, S3 and S4). We used five different independent images from the Z-stack and analyzed thousands of mitochondria association condition and performed statistical analysis (Figure 5E). In our statistical analysis, we found that the mitochondria associations were significantly decreased in the mutant lines, while this could be complemented with the *ipgam1/2* double mutant complemented by native promoter with nonfunctional phosphoglycerate mutase. (line 231-235).

The concerns are the mutants are retarded in growth compared to the wild type – Supplemental Figure S7 and Figure S8. This severe growth phenotype will have significant impact on many aspect in the cell – does it affect mitochondrial mass, morphology, number (I realise all are linked), and for chloroplast are they affected in mass, size etc. Does it affect cell size, cell division etc.

Response:

In the enolase mutant, it has been reported that both cell shape or cell size may be altered (Eremina et al., 2015). To avoid the side effect of changed cell shape and size, we complemented the *ipgam1/2* double mutant using a native promoter driven nonfunctional phosphoglycerate mutase. The resultant genotype displays a similar whole plant phenotype to the double mutant, however, the mitochondrial association with chloroplasts is increased (Figure 6). Furthermore, the nuclear sublocalized enolase resulted in a complementation of the enzyme activity without complementing the formation of the glycolytic metabolon, the plant growth phenotype (Eremina et al., 2015; Kang et al., 2013), or the cell biological phenotype (Figures 5D and E). Moreover, to further answer the reviewer's concern, we undertook an independent experiment. The mitochondrial GFP lines were treated with an enolase inhibitor (ENOblock hydrochloride). This inhibited the enzyme activity and induced enolase nuclear translocation, where the enzyme is thought to act as a transcriptional repressor (Cho et al., 2017). ENOblock is the first reported enolase inhibitor which is a non-substrate analogue that directly binds to enolase and inhibits its activity. The associations and movement of mitochondria to chloroplast were significantly decreased following 40 min treatment by ENOblock in comparison to the mock control (Figure 7). Given that the ENOblock induced relocalisation of the enolase to the nucleus, we assume that the formation of the metabolon at outer mitochondria membrane is significantly decreased, thus the associated mitochondria to chloroplast are also significantly decreased. The plant phenotype, cell size, mitochondria size and chloroplast size were not changed following this short 40 mins treatment. These combined results provide strong support for the hypothesis raised in the previous submission. (line 244-251)

Figure 6 – Agreed that mobility is affected, this relates to Figure 5 and 6 it appears that the morphology of the phosphoglycerate mutant (but not the enolase mutant) is greatly affected – given the proteins involved in mitochondrial fusion and fission bind to mitochondria (and chloroplast, and peroxisomes), have the mutants impacted this association and the differences observed are secondary in nature. Also how is binding of mitochondria to the components of the cytoskeleton affected.

Response:

Yes, the plant phenotype of *ipgam1/2* mutant is significantly changed compared with the WT. To rule out the possibility that the mitochondria movement and association effected by this phenotype, we complemented the *ipgam1/2* mutant by nonfunctional phosphoglycerate mutase. It could complement the protein complex while the enzyme activity of glycolysis is still missing. Thus, the plant phenotype is not changed, while the mitochondria association are significantly increased (Figure 6c, e and 7c, e). This result proves that the mitochondria and chloroplast association is affected by the lower levels of the glycolytic metabolon. In addition, we crossed the mutant of *pgam1/2* double mutant with the actin marker but this was not been successful because of the very weak plant growth phenotype of the *pgam1/2* mutant. However, we additionally treated actin marker lines with Enoblock and its mock control. In this experiment actin was unaffected

while the association of the mitochondria to chloroplast was again significantly affected. A comment to this affect was added to the revised manuscript. (line 244-254)

Thus, overall what cannot be ruled out, and is a possibility is that the cell biology of these mutants is radically altered, as would be expected from such a fundamental pathways being disrupted.

Independent lines of evidence are needed – as the authors obtained for the interaction data.

Response:

We used the mitochondrial GFP lines and treated with enolase inhibitor (AP-III-a4 hydrochloride). Following these inhibitor treatments, the cellular and chloroplast are unchanged but the association of mitochondria to chloroplast significantly decrease (Figure 7). This result shows that altered association is independent of cellular morphology and thus strongly supports our hypothesis.

To support the claims that the author state there needs to be independent evidence that would rule out or at least make it much less likely. How the authors would like to do this is up to them but what would convince me is to have a mutant that has lost activity (point mutation in the active site) and carry out this study. That way the enzyme is there but the activity isn't. Or the converse have the activity without the domain that interacts with the other enzymes. Other approaches is detailed EM work with immunolabelling. Biochemical approaches include cross-linking to pull down these structures intact.

Response:

We already used the *sdmA-pgam1/2* lines which is the double mutant complemented by native promoter with nonfunctional phosphoglycerate mutase. The PGAM-SDAM lines could complement the mitochondria association (Figure 6C and I). In addition to reviewer's concern, we processed the enolase inhibitor experiment as mentioned above. The plant phenotype, cell size, mitochondria size and chloroplast size are not changed in the 40 mins short treatment. This result strongly supports our hypothesis.

Reviewer #3 (Remarks to the Author):

This manuscript provides compelling evidence for both metabolite channelling between the glycolytic enzymes PGM and enolase (by using stable isotope labelling to demonstrate that the substrates have decreased interaction with the bulk phase). Critically, they also show that these enzymes are also involved in a physical interaction

between chloroplasts and mitochondria. Together, these observations provide a very important advance in knowledge about the spatial organisation of metabolism. I do not have any major criticisms of the methods or interpretation of the results.

I have a few minor comments that the authors may wish to address.

Lines 64-65. It is not clear what point is being made and it is not immediately obvious from the references. I am wondering if this is a reference to the “chemotactic” movement of enzymes along substrate gradients- which has been proposed but could be an artifact of the measurement method (discussed in Smirnov, 2019).

Response:

We have improved it accordingly (line 65-68).

Line 218. “hypothesis” might be better than “theory” in the context.

Response:

We changed it in text.

Lines 343-345. I am not sure that the resolution of microscopy used here can provide information on the size of protein complexes that could bridge or attach the mitochondria and chloroplasts.

Response:

We proposed that the protein complex may bridge the mitochondria and chloroplast while the resolution of current confocal microscopy is still very difficult to find the bridge between mitochondria and chloroplast.

Figure 5 and 6 legends. It would help to identify all the abbreviations of enzyme constructs used in the figures.

Response:

We added this information.

- Achnine, L., Blancaflor, E. B., Rasmussen, S., & Dixon, R. A. (2004). Colocalization of L-phenylalanine ammonia-lyase and cinnamate 4-hydroxylase for metabolic channeling in phenylpropanoid biosynthesis. *The Plant Cell*, *16*(11), 3098-3109.
- Bar-Even, A., Flamholz, A., Noor, E., & Milo, R. J. B. e. B. A.-B. (2012). Thermodynamic constraints shape the structure of carbon fixation pathways. *1817*(9), 1646-1659.
- Cho, H., Um, J., Lee, J.-H., Kim, W.-H., Kang, W. S., Kim, S. H., . . . Jung, D.-W. (2017). ENOblock, a unique small molecule inhibitor of the non-glycolytic functions of enolase, alleviates the symptoms of type 2 diabetes. *Scientific reports*, *7*, 44186.
- Eremina, M., Rozhon, W., Yang, S., & Poppenberger, B. (2015). ENO2 activity is required for the development and reproductive success of plants, and is feedback-repressed by AtMBP-1. *The Plant Journal*, *81*(6), 895-906.
- Garrett, R. G., C. M. (2005). . . (2005). *Biochemistry (3rd ed.)*. Belmont, CA: Thomson Brooks/Cole.
- Graham, J. W., Williams, T. C., Morgan, M., Fernie, A. R., Ratcliffe, R. G., & Sweetlove, L. J. (2007). Glycolytic enzymes associate dynamically with mitochondria in response to respiratory demand and support substrate channeling. *Plant Cell*, *19*(11), 3723-3738.
- Hooper, C. M., Castleden, I. R., Tanz, S. K., Aryamanesh, N., & Millar, A. H. J. N. a. r. (2017). SUBA4: the interactive data analysis centre for Arabidopsis subcellular protein locations. *45*(D1), D1064-D1074.
- Islam, M. S., & Takagi, S. (2010). Co-localization of mitochondria with chloroplasts is a light-dependent reversible response. *Plant signaling & behavior*, *5*(2), 146-147.
- Jacobson, T. B., Korosh, T. K., Stevenson, D. M., Foster, C., Maranas, C., Olson, D. G., . . . Amador-Noguez, D. J. M. (2020). In Vivo Thermodynamic Analysis of Glycolysis in *Clostridium thermocellum* and *Thermoanaerobacterium saccharolyticum* Using ¹³C and ²H Tracers. *5*(2).
- Kang, M., Abdelmageed, H., Lee, S., Reichert, A., Mysore, K. S., & Allen, R. D. (2013). AtMBP-1, an alternative translation product of LOS2, affects abscisic acid responses and is modulated by the E3 ubiquitin ligase AtSAP5. *The Plant Journal*, *76*(3), 481-493.
- Laursen, T., Borch, J., Knudsen, C., Bavishi, K., Torta, F., Martens, H. J., . . . Bassard, J. E. (2016). Characterization of a dynamic metabolon producing the defense compound dhurrin in sorghum. *Science*, *354*(6314), 890-893. doi:10.1126/science.aag2347
- Noor, E., Bar-Even, A., Flamholz, A., Reznik, E., Liebermeister, W., & Milo, R. J. P. C. B. (2014). Pathway thermodynamics highlights kinetic obstacles in central metabolism. *10*(2), e1003483.
- Oikawa, K., Matsunaga, S., Mano, S., Kondo, M., Yamada, K., Hayashi, M., . . . Higashi, S. (2015). Physical interaction between peroxisomes and chloroplasts elucidated by in situ laser analysis. *Nature Plants*, *1*(4), 35-42.
- Pareek, V., Tian, H., Winograd, N., & Benkovic, S. J. (2020). Metabolomics and mass spectrometry imaging reveal channeled de novo purine synthesis in cells. *Science*, *368*(6488), 283-290.
- Park, J. O., Rubin, S. A., Xu, Y.-F., Amador-Noguez, D., Fan, J., Shlomi, T., & Rabinowitz, J. D. J. N. c. b. (2016). Metabolite concentrations, fluxes and free energies imply efficient enzyme usage. *12*(7), 482.
- Park, J. O., Tanner, L. B., Wei, M. H., Khana, D. B., Jacobson, T. B., Zhang, Z., . . . Stevenson, D. M. J. N. c. b. (2019). Near-equilibrium glycolysis supports metabolic homeostasis and energy yield. *15*(10), 1001-1008.
- Ruberti, C., Barizza, E., Bodner, M., La Rocca, N., De Michele, R., Carimi, F., . . . Zottini, M. (2014). Mitochondria change dynamics and morphology during grapevine leaf senescence. *PLoS One*, *9*(7).

- Sharma, M., Bennewitz, B., & Klösgen, R. B. (2018). Dual or not dual?—Comparative analysis of fluorescence microscopy-based approaches to study organelle targeting specificity of nuclear-encoded plant proteins. *Frontiers in Plant Science*, *9*, 1350.
- Shearer, G., Lee, J. C., Koo, J. a., & Kohl, D. H. J. T. F. j. (2005). Quantitative estimation of channeling from early glycolytic intermediates to CO₂ in intact *Escherichia coli*. *272*(13), 3260-3269.
- Wheeldon, I., Minter, S. D., Banta, S., Barton, S. C., Atanassov, P., & Sigman, M. (2016). Substrate channelling as an approach to cascade reactions. *Nature chemistry*, *8*(4), 299.
- Zhang, Y. J., Beard, K. F. M., Swart, C., Bergmann, S., Krahnert, I., Nikoloski, Z., . . . Obata, T. (2017). Protein-protein interactions and metabolite channelling in the plant tricarboxylic acid cycle. *Nature Communications*, *8*. doi:10.1038/ncomms15212

REVIEWERS' COMMENTS:

Reviewer #1 (Remarks to the Author):

The authors have addressed a number of comments raised by reviewers, unfortunately as to why the channel exists is a remaining curiosity. Four possible rationales are given to explain the value of a channel: regulation, cytotoxic metabolites, unstable metabolites, or leakage that impacts the cellular operational efficiency. While this is a good description of the possibilities, none of these is particularly convincing. First, the reason to have the two reactions linked to the pyruvate kinase reaction is because they are unfavorable and pyruvate kinase pulls through, linkage of the first 2 has no thermodynamic value. Also these metabolites are not believed to be toxic or unstable and their leakage impacting the cell is also unlikely. Perhaps descriptions of in vivo and in vitro delta Gs are not congruent, which the authors suggest as well? There can certainly be important differences between in vivo and in vitro results, however the delta Gs are described in many systems, if not in plants, then more generally in others – it seems unlikely that the case would be so different for plants than other systems in this most basic central pathway? Pertaining to the response that reactions 8 and 9 be coupled but not to 10, seems to have lost me, the entire reason 8 and 9 become feasible so through linkage to 10 which pulls them through. I would think that the authors at a minimum would want to acknowledge this point fully to show it was given full consideration and remains a quandary?

Reviewer #2 (Remarks to the Author):

I Have carefully reassessed and review this revised version of the manuscript and based on two points I think the conclusions are now supported by a variety of different approaches.

1) I mis-understood the nature of the mutant in Figure 6 as a deletion mutant rather than a non-functional enzyme – thus I requested this as an additional experiment. This is strong supporting evidence for the associations as concluded in the manuscript. I thank the authors for pointing this out, and can see this now both in the results and discussion.

2) The addition of new independent evidence using the inhibitor provides very strong support to the hypothesis. Given that it is a completely different approach IT provides another level of confidence that the interpretations are reasonable.

Reviewer #3 (Remarks to the Author):

The authors have addressed the few queries that I had. Indeed, given the immense technical difficulties, I believe they have amassed a good amount of evidence to support their conclusions. However, in responding to other reviewers, they have included new experiments with ENOBlock, which they used as inhibitor of enolase. The results supported their conclusions about the role of the metabolon in mito-chloroplast association. However, I was unable to find a reference in the revised manuscript that would lead the reader to relevant published evidence for its mode of action. I found one reference that casts doubt on its mode of action (ENOblock Does Not Inhibit the Activity of the Glycolytic Enzyme Enolase. <https://dx.plos.org/10.1371/journal.pone.0168739>). This might need a comment.

REVIEWERS' COMMENTS:

Reviewer #1 (Remarks to the Author):

The authors have addressed a number of comments raised by reviewers, unfortunately as to why the channel exists is a remaining curiosity. Four possible rationales are given to explain the value of a channel: regulation, cytotoxic metabolites, unstable metabolites, or leakage that impacts the cellular operational efficiency. While this is a good description of the possibilities, none of these is particularly convincing. First, the reason to have the two reactions linked to the pyruvate kinase reaction is because they are unfavorable and pyruvate kinase pulls through, linkage of the first 2 has no thermodynamic value. Also these metabolites are not believed to be toxic or unstable and their leakage impacting the cell is also unlikely. Perhaps descriptions of in vivo and in vitro delta Gs are not congruent, which the authors suggest as well? There can certainly be important differences between in vivo and in vitro results, however the delta Gs are described in many systems, if not in plants, then more generally in others – it seems unlikely that the case would be so different for plants than other systems in this most basic central pathway? Pertaining to the response that reactions 8 and 9 be coupled but not to 10, seems to have lost me, the entire reason 8 and 9 become feasible so through linkage to 10 which pulls them through. I would think that the authors at a minimum would want to acknowledge this point fully to show it was given full consideration and remains a quandary?

Response: Kinetic characterization of this metabolon indicates that it was considerably more efficient than the free enzymes of the extra-plastidial preparation (exhibiting approximately five times higher substrate affinity and 19 times higher efficiency, Figure 4d). Given that the (dis)assembly of glycolytic metabolon appears to depend on the energy needs of the cell (Graham et al., 2007), future research should also study if the (dis)assembly of the metabolon are dependent on the in vivo energy requirements of the cell. The exact benefit of the metabolon is currently hard to distinguish. We agree that while four possible rationales are given to explain the value of a channel which one applies to the metabolon here is unclear it is possible that there is a thermodynamic benefit. The metabolic intermediates are not believed to be toxic or unstable and their leakage impacting the cell is also unlikely. Moreover, looking at the in vivo thermodynamics of these reactions in other systems (Jacobson et al., 2020), data is not available to estimate these in plants, suggests that there is likely no thermodynamic advantage to metabolon formation. Thus, despite careful consideration the exact purpose of the metabolite channel currently remains opaque and will require further study to clarify.

Reviewer #2 (Remarks to the Author):

I Have carefully reassessed and review this revised version of the manuscript and based on two points I think the conclusions are now supported by a variety of different approaches.

1) I mis-understood the nature of the mutant in Figure 6 as a deletion mutant rather than a non-functional enzyme – thus I requested this as an additional experiment. This is strong supporting evidence for the associations as concluded in the manuscript. I thank the authors for pointing this out, and can see this now both in the results and discussion.

2) The addition of new independent evidence using the inhibitor provides very strong support to the hypothesis. Given that it is a completely different approach IT provides another level of confidence that the interpretations are reasonable.

Response:

Thanks for the nice comments.

Reviewer #3 (Remarks to the Author):

The authors have addressed the few queries that I had. Indeed, given the immense technical difficulties, I believe they have amassed a good amount of evidence to support their conclusions. However, in responding to other reviewers, they have included new experiments with ENOBlock, which they used as inhibitor of enolase. The results supported their conclusions about the role of the metabolon in mito-chloroplast association. However, I was unable to find a reference in the revised manuscript that would lead the reader to relevant published evidence for its mode of action. I found one reference that casts doubt on its mode of action (ENOblock Does Not Inhibit the Activity of the Glycolytic Enzyme Enolase. <https://dx.plos.org/10.1371/journal.pone.0168739>). This might need a comment.

Response: Thanks for the nice suggestion.

Currently the mode-of-action of ENOBlock is not unambiguously defined. However, as the reviewer states this does not affect the active site of the enzyme but rather its binding and subsequent subcellular localization a comment to this effect has been added to the manuscript.

Graham, J. W. A., Williams, T. C. R., Morgan, M., Fernie, A. R., Ratcliffe, R. G., & Sweetlove, L. J. (2007). Glycolytic enzymes associate dynamically with mitochondria in response to respiratory demand and support substrate channeling. *Plant Cell*, 19(11), 3723-3738.
doi:10.1105/tpc.107.053371

Jacobson, T. B., Korosh, T. K., Stevenson, D. M., Foster, C., Maranas, C., Olson, D. G., . . . Amador-Noguez, D. J. M. (2020). In Vivo Thermodynamic Analysis of Glycolysis in *Clostridium thermocellum* and *Thermoanaerobacterium saccharolyticum* Using ¹³C and ²H Tracers. 5(2).